# Coherence requirements for quantum communication from hybrid circuit dynamics

Shane P. Kelly,[1, 2, *] Ulrich Poschinger,[1] Ferdinand Schmidt-Kaler,[1] Matthew P.A. Fisher,[3] and Jamir Marino[1, 2]

[1]*Institut für Physik, Johannes Gutenberg Universität Mainz, D-55099 Mainz, Germany*
[2]*Kavli Institute for Theoretical Physics, University of California, Santa Barbara, CA 93106-4030, USA*
[3]*Department of Physics, University of California, Santa Barbara, CA 93106-4030, USA*
(Dated: October 31, 2023)

The coherent superposition of quantum states is an important resource for quantum information processing which distinguishes quantum dynamics and information from their classical counterparts. In this article we determine the coherence requirements to communicate quantum information in a broad setting encompassing monitored quantum dynamics and quantum error correction codes. We determine these requirements by considering hybrid circuits that are generated by a quantum information game played between two opponents, Alice and Eve, who compete by applying unitaries and measurements on a fixed number of qubits. Alice applies unitaries in an attempt to maintain quantum channel capacity, while Eve applies measurements in an attempt to destroy it. By limiting the coherence generating or destroying operations available to each opponent, we determine Alice's coherence requirements. When Alice plays a random strategy aimed at mimicking generic monitored quantum dynamics, we discover a coherence-tuned phase transitions in entanglement and quantum channel capacity. We then derive a theorem giving the minimum coherence required by Alice in any successful strategy, and conclude by proving that coherence sets an upper bound on the code distance in any stabilizer quantum error correction code. Such bounds provide a rigorous quantification of the coherence resource requirements for quantum communication and error correction.

## I. INTRODUCTION

Protecting quantum superposition is essential for obtaining quantum advantage in simulation, sensing, communication and computation. While noisy intermediate-scale quantum devices have advanced this frontier [1–3] by improving the fidelities of quantum gates and qubits, quantum error correction is conjectured to be essential in the long run. Similar to classical error protection, quantum error correction requires redundancy in the number of qubits and other quantum resources such as entanglement. Thus, there has been a significant effort towards developing quantum resource theories [4], which rigorously determine the nature and quantity of a given quantum resource such as entanglement [5, 6], non-locality [7, 8], or quantum coherence [9–13] (related to superposition [14, 15]). For example, entanglement and coherence have been demonstrated as essential resources for performing quantum sensing [12, 16, 17]. At the same time, any resource for any given quantum resource theory is useful in some channel discrimination task [18–24]. Finally, Ref. [25] provided an error correction protocol that consumes coherence as it corrects errors. While this protocol shows that coherence can be used as a resource for error correction, it is not yet known if or how coherence is necessary for quantum communication.

In this article, we determine the coherence resource requirements for quantum communication in a generic setting encapsulating both monitored many body quantum dynamics and error correction protocols involving active feedback. In the first case, we investigate an en-

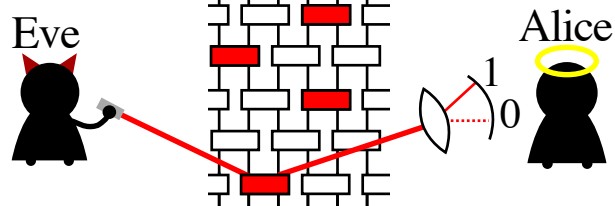

Figure 1. Cartoon hybrid circuit composed of unitaries (white) and measurements (red), which can be viewed as an information game played between Alice and Eve. In this game, Alice attempts to store her diary in a set of qubits and Eve attempts to destroy her diary by measuring them (for instance, shining light on the qubits). To protect her diary, Alice applies a set of unitaries in an attempt to protect her diary from Eve's measurements, which Eve can apply at a fixed rate relative to the rate at which Alice can apply unitaries. Alice is allowed to capture the emitted light, record the measurement outcomes, and knows the measurement basis Eve makes a measurement in. In this way, Alice can keep track of the evolving state of the system, such that she might be able to apply a unitary at the end of the game to recover her diary. The measurement-induced phase transition occurs when random strategies are played, and corresponds to a transition in the quantum channel capacity between the initial state encoding Alice's diary, and the final state at the end of the game. Alice wins the game in the volume-law phase when the quantum channel capacity is finite, while she loses the game in the area-law phase when the channel capacity is zero.

semble of hybrid circuits modeling a generic class of monitored quantum dynamics, and identify phase transitions in their channel capacity [26–30]. By controlling the coherence generating capacity of such circuits we are able to control the phase transition and extract the coherence requirements for obtaining a finite channel capacity.

* Corresponding author: shakelly@uni-mainz.de

The hybrid circuit channels previously studied [31–71] are composed of a sequence of random local unitaries and measurements which compete to drive the phase transition in the channel capacity and entanglement properties. While the transition was first observed in the scaling properties of entanglement [32–35], the manifestation of the transition in terms of channel capacity allows us to investigate the information game shown in Fig. 1. In this game, an agent Alice, having access to a set of qubits and a limited set of local unitary operations, attempts to protect a quantum diary (i.e. an arbitrary quantum state) from Eve who attempts to destroy it with the application of quantum measurements (for which Alice can record the outcomes of). Alice wins, if at the end of the game there is a finite quantum channel capacity and she is able to recover her diary, while she looses when the channel capacity is zero.

By limiting the coherence generating ability of Alice, we are able to identify the coherence resources required by Alice to protect her quantum diary. To identify which operations are coherence generating and which are destroying, we make use of the resource theory of coherence [10], which is a basis specific resource theory, that quantifies the amount of quantum superposition in that basis. By using the relative entropy of coherence [9] as the resource quantifier, and considering its dynamics in the information game played between Alice in Eve, we uncover the coherence resources required by Alice to protect both classical and quantum diaries.

Inspired by these results, we apply the intuition and tools developed studying the information game to quantum error correction and find that the code distance of a stabilizer quantum error correction code is bounded by the relative entropy of coherence in any basis.

### A. Coherence requirements in communication games and quantum error correction

We first consider the limit of zero coherence, and confirm Alice can only protect classical information. That is, we show that if Alice can only prepare coherence-free states, and only perform the free operations of the coherence resource theory, then she can only store and protect classical information from a set of errors introduced by Eve. If instead, Alice is given a state (with coherence) encoding a quantum diary, then she can protect quantum information using coherence non-generating unitaries (the free operations of the coherence resource theory) as long as the measurements Eve performs in her attack are restricted to preserve coherence. Next, we allow Alice to generate coherence and identify the minimum amount of coherence she must maintain to preserve her quantum diary.

In this setting, we first consider the case in which Alice and Eve take random strategies in order to sample the coherence requirements for a large class of monitored quantum dynamics. The first set of strategies result

in a hybrid circuit composed of random controlled not gates (CNOTs) and projective measurements which can either generate or destroy coherence. This investigation finds that, even at arbitrarily weak measurements, Alice's ability to protect quantum information can undergo a transition tuned by the relative rate of coherence destroying and generating measurements. Next, we allow Alice to use a limited rate, $p_R$, of coherence generating unitaries in her random strategy and identify the threshold rate of decohering measurements, $p_m^c$, below which she can protect her quantum diary. We find that the threshold rate of decohering measurements $p_m^c$ increases linearly with $p_R$ (i.e. $p_m^c = \alpha p_R$ for some constant $\alpha$), indicating a greater capacity to protect her quantum diary given access to more coherence generating unitaries. Finally, we identify bounds on the minimum coherence Alice must preserve in her qubits under any strategy taken by her or Eve. While the phase transition occurring in random circuits demonstrates that coherence is a requirement for quantum communication in monitored many body quantum dynamics, this bound provides a rigorous quantification of that requirement.

This suggests that one could also identify a threshold amount of coherence required for error correction. Indeed, we find such a relation between the relative entropy of coherence (computed for a specific state in the quantum code space) and the code distance (the number of errors correctable by the stabilizer code). A weak formulation of this relation is that the relative entropy of coherence of the maximum coherent state in the code space bounds the code distance (Theorem 3 in Sec. V). A stronger formulation can be obtained by considering subspaces of the full code space since a bound on the code distance for a subspace is a bound for the full code space. Using one such stronger constraint on the code distance, we show that our bound reproduces the classical Singleton bounds when applied to Calderbank-Shor-Steane (CSS) codes which are a type of quantum error correction code constructed from two classical error correction codes [72–74].

Using our bound, we therefore provide a rigorous quantification of the coherence resource requirements for constructing a quantum error correction code. Intuitively, this bound gives the extra coherent resources required to encode a quantum state. While it is natural that the coherence of the physical state must be greater than that of the encoded state, our bound shows that the coherence is also constrained by the number of errors that one desires to correct. Thus, it gives the amount of extra coherence required for error correction than required to simply represent the state.

We begin in section II by reviewing the resource theory of coherence and one of its resource quantifiers, the relative entropy of coherence. Then, we introduce the information game and monitored quantum dynamics and discuss the unitary limit of such models. Then, in section III, we discuss the coherence-free limit, and show that Alice can only encode and protect classical infor-

mation in this limit. We elaborate on this result in sections III B and III C by discussing Alice's ability to protect quantum information only using coherence non-generating operations. In section IV we investigate the dynamics of coherence induced by measurements and show that, while Eve can always destroy Alice's diary if she is restricted to using coherence non-generating unitaries, Alice can protect a quantum diary if Eve accidentally generates coherence by performing measurements in the wrong basis. Finally in section V we discuss the coherence resource requirements for quantum communication: both the requirements for Alice to protect her diary (section V A), and the requirements in quantum error correction code design (section V B).

### B. New perspectives on hybrid circuit dynamics

While our work primarily focuses on using hybrid circuit dynamics to understand the coherence resource requirements for quantum communication, it also provides perspectives and results interesting for the reader primarily interested in hybrid circuits dynamics and their transitions.

**Classical circuits** The first set of hybrid circuits we consider are similar to the classical dynamics discussed in Refs. [75–78] which show measurement-induced transitions in classical information and chaos quantifiers. Similarly, we first consider in section III A, the dynamics of a circuit composed of random CNOTs occurring at a fixed rate and random bit erasers occurring at a rate, $p_e$, relative to the rate of CNOTs, and find a classical purification transition. The classical purification transition is investigated by considering the dynamics of an initial classical distribution of classical bit strings and observing that the late time entropy of that distribution undergoes a phase transition similar to the quantum purification transitions discussed in Refs. [26]. Going beyond previous works, we show that this transition in a classical entropy is the coherence-free limit of a more general class of dynamics. We then show in section III B that if the initial state has quantum coherence, and the bit erasers are implemented using a sequence of measurements that preserve coherence, then the transition can also be considered as a quantum purification transition. We show in section III C that the dynamics of the circuit can be described by the dynamics of a classical code space.

**Coherence controlled entanglement transitions** The second set of hybrid circuits we consider are composed of CNOTs and measurements which occur at a relative rate, $p_m$, to the CNOTs. We then randomly choose to measure in the $X$, $Y$ or $Z$ bases with probability $p_x$ $p_y$ and $p_z = 1 - p_x - p_y$ respectively. For these circuits, the dynamics of the relative entropy of coherence in the $X$ and $Z$ basis are particularly interesting because they constrain the amount of entanglement in the system (see section II D). Furthermore, the dynamics of coherence are analytically accessible both in the measurement-only

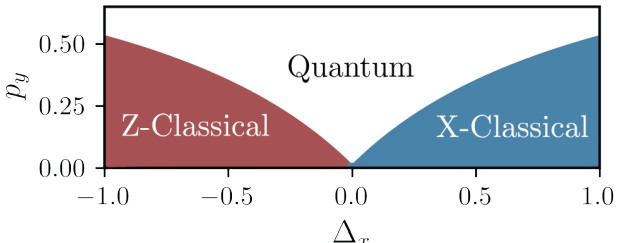

Figure 2. Sketch phase diagram for a circuit composed of CNOTs, and vanishing rate of measurements $p_m$. In this circuit, we randomly choose to measure in the $X$, $Y$ or $Z$ measurements with probability $p_x$ $p_y$ and $p_z = 1 - p_x - p_y$ respectively (we define $\Delta_x = (p_x - p_z)/(1 - p_y)$). The phases labeled $Z$-classical and $X$-classical are area-law phases in which superposition is vanishing in the $Z$ and $X$ basis respectively. Typical states in these phases are therefore efficiently represented by a superposition of a limited number of $X$ or $Z$ basis states. This is in contrast to the "quantum" phase which shows volume-law entanglement, and is characterized by a large amount of superposition as quantified by the relative entropy of coherence, both in the $X$ and $Z$ basis.

limit $p_m \to \infty$ and vanishing measurement rate limit $p_m \to 1/L^2$. In the first case, the coherence can be predicted exactly, but the entanglement dynamics are trivial and the steady states are all product states.

In the second limit, of vanishing measurement rate, an entanglement transition can be observed as a function of the relative probabilities of which Pauli operators are measured, $p_y$ and $\Delta_x = (p_x - p_z)/(1 - p_y)$. In this limit, the dynamics of coherence follow a Markov process described by a random walk in the amount of information 'known' about the $X$ and $Z$ basis states (see Fig. 7). By studying this walker we find that the superposition (coherence) in the $X$ basis increases at a rate $p_y + p_z - p_x$. Thus, if $p_x > p_y + p_z$, the amount of superposition in the $X$ basis vanishes, the state becomes classical in the $X$ basis with no entanglement. Similarly in the $Z$ basis, if $p_z > p_x + p_y$, the superposition in the $Z$ basis vanishes, the state becomes classical in the $Z$ basis and entanglement is again not allowed to form. Thus, if we consider the entropy of a state where we first take the infinite time limit and then the vanishing measurement rate limit $\lim_{p_m \to 0} \lim_{t \to \infty} S$, we find a transition in the entanglement $S$ at the critical line $|p_x - p_z| > p_y$. This is summarized in Fig. 2, where we have a phase transition between states classical in the $X$ and $Z$ basis, and quantum states with volume-law entanglement.

The random walk describing the dynamics of coherence is discussed in section IV along with the coherence controlled entanglement transition. Details about the Markovian dynamics of coherence are given in appendix D.

**What is quantum about the volume-law entangled phases?** In the final section V we find that there is a transition in the ability of Alice to protect a quantum

code, controlled by the rate at which she can generate coherence. In that section, Eve makes an attack with coherence preserving bit erasers and a coherence destroying measurement, while Alice defends her diary with CNOTs and a limited rate of phase gates. While in that section, we only discuss her ability to protect a quantum diary, we show in section VI that there is a transition between a regime where Alice can only protect a classical diary to one where she can protect a quantum diary. The difference between the two phases provides an answer to what is quantum about the entangling phase of the measurement-induced transitions discussed in Refs. [32–35] in comparison to the classical transitions discussed in Refs. [75–78]. Here, the transition between being able to protect a classical diary to being able to protect a quantum diary occurs as the ability to correct both bit and phase errors as apposed to just bit errors. Thus, this answer to what is quantum about the entangling phase of the measurement-induced phase transitions is equivalent to the answer to what is quantum about quantum error correction codes [72, 79]. While classical error correction and the classical scrambling phases protect information from just bit or just phase errors, the entangling phase and quantum error correction codes are robust to both bit and phase errors.

## II. COHERENCE IN MONITORED QUANTUM DYNAMICS

### A. Resource theory of coherence

In this paper, we determine how quantum error correction requires quantum superposition and quantify exactly how much superposition is required. We do this using the quantum resource theory of quantum coherence [9, 10], which we will outline in this section. We also discuss in depth the relative entropy of coherence which provides an important and intuitive quantification of coherent superposition.

Resource theories [4] provide a formal setting by identifying a set of resource free states (product states for the resource theory of entanglement), a set of free operations (i.e. Local Operations and Classical Communication ), and then asks what operations and tasks are made available with the possession of a resource state (i.e. an entangled state). The resource theory of coherence aims to quantify the resourcefulness of a state with coherent superposition in a given basis, and thus there is a different resource theory of coherence for each basis $D$. The free states for the coherence in a basis $D$ are the set of diagonal states satisfying $\rho = \rho_D \equiv \sum_{d \in D} \rho_{dd} |d\rangle \langle d|$ where $\{|d\rangle\}$ are the basis states of the basis $D$. Importantly, this set of free states can not contain any pure states with quantum superposition in $D$ since such states would have off diagonal terms.

Similar to the freedom in choosing the free operations of an entanglement resource theory (local unitaries v.s.

Local Operations and Classical Communication), the resource theory of coherence also has multiple choices of free operations [13]. In this work, we limit our considerations to the free operations introduced in Ref. [9] called *"Incoherent Operations"*, therein defined as the set of quantum channels where each Krauss operator of the quantum channel takes diagonal states to diagonal states: $E_k \rho_1 E_k = \rho_2$ where it is required that $\rho_2$ is diagonal in the basis $D$ if $\rho_1$ is. Such a constraint ensures that states with superposition can not be created by the set of Incoherent Operations, while at the same time is loose enough to allow maps between different basis states as required for classical computation. This is in contrast to other choices of free operations [10, 13] such as strictly Incoherent Operations [80] which do not allow for classical computations.

#### 1. Relative entropy of coherence

Similar to how the resource theory of entanglement allows for a multitude of entanglement monotones (i.e. Log-negativity, relative entropy of entanglement, . . . ), the resource theory of coherence also has a multitude of resource quantifiers [4]. Here, we only consider the relative entropy of coherence because it provides an intuitive quantification for the amount of coherent superposition possessed by a given state. The relative entropy of coherence $C(\rho, D)$ in a basis, $D$, is defined for a state $\rho$ as:

$$C(\rho, D) = S(\rho_D) - S(\rho), \tag{1}$$

where $S(\rho) = -tr[\rho \log \rho]$ is the von Neumann entropy of a mixed state $\rho$ and $\rho_D$ is again the diagonal part of $\rho$ in the basis $D$. In this work, we will focus on the Hilbert space of $L$ qubits and consider two coherence resource theories: one in the computation basis ($D = X$ basis) with basis states $|x\rangle = |x_1, x_2, \ldots x_L\rangle$ and for which the Pauli $X_i$ operators are diagonal, $X_i |x\rangle = (-1)^{x_i} |x\rangle$, and one (the $D = Z$ basis) with basis states $|z\rangle = |z_1, z_2, \ldots z_L\rangle$ in which the $Z_i$ Pauli operators are diagonal. Below, we will refer to the relative entropy of coherence simply as coherence $C(\rho, D)$, and label these coherences in the $D = X$ and $D = Z$ bases as $C_x = C_x(\rho) = C(\rho, X)$ and $C_z = C_z(\rho) = C(\rho, Z)$ where the state is often implied by context.

For pure states, the coherence is equivalent to the Shannon entropy of the probability distribution for the measurement results $P(d) = \langle d| \rho |d\rangle$ pertaining to the basis $D$. Explicitly, $C(\rho, D) = H(P(d))$ where $H(P)$ is the Shannon entropy of a distribution $P$. Thus, the coherence of a pure state is the amount of statistical entropy over which basis states the quantum state $|\psi\rangle$ is a superposition in. For example, a single qubit polarized in the $+Z$ direction is an equal superposition of two $X$ basis states, and thus has coherence $C_x = 1$, while a pure state polarized in the $X$ direction has zero coherence in

| Who can apply $X$-coherence generating operations? | Section | The operation |
|---|---|---|
| Neither Alice nor Eve | III | N/A |
| Eve only | IV | $Y$ and $Z$ measurements |
| Alice only | V | Phase gates |

Table I. Table showing which agent can apply the listed $X$-coherence generating operations in the various sections of the paper. All other circuit operations occurring in the hybrid circuit considered do not generate coherence in the $X$ basis and are therefore Incoherent Operations in the $X$ basis (free operations of the $X$ basis coherence resource theory).

the $X$ basis $C_x = 0$. For mixed states, consider the example of a product state of $N_x$ bits polarized in the $X$ basis, $N_z$ bits polarized in the $Z$ basis, and $M$ completely mixed bits each in a state $\rho_i = (|0\rangle \langle 0| + |1\rangle \langle 1|)/2$. Such a state has $S(\rho) = M$ due to the $M$ completely mixed bits, and $H(P(x)) = N_z + M$ since both the $Z$ polarized bits and the completely mixed bits are completely uncertain about the $X$ basis states. Thus the coherence in the $X$ basis is $C_x(\rho) = N_z$.

### B. Monitored quantum dynamics as an information game

In this work, we investigate an information game (see Fig. 1) to determine the coherence resource requirements for quantum communication in monitored quantum dynamics. This game involves two opponents, Alice and Eve, who compete by applying random unitary and measurements in an attempt to maintain or destroy the quantum channel capacity of the resulting hybrid circuit (see Fig. 1). Alice attempts to store and maintain a diary in the evolving set of qubits by applying random unitaries, and wins when the resulting hybrid circuit has a finite quantum channel capacity. While Eve attempts to destroy the diary by applying local Pauli measurements and wins when the resulting hybrid circuit has zero quantum channel capacity. By restricting the coherence resource generating or destroying operations of each opponent, we learn about the coherence requirements for quantum communication by observing who wins the information game.

A sensible study of a resource theory first starts with an investigation to what the resource free states and operations can accomplish, and then studies the additional tasks achievable with the aid of various resource generating operations. Our investigation follows this approach guided by the coherence resource theory defined with the $X$ Pauli basis: first in section III, we consider games in which Alice and Eve are restricted to only perform Incoherent Operations; then in section IV, we allow Eve to perform coherence generation operations; and then in section V we also allow Alice to perform coherence generation operations.

By considering the set of all possible strategies Alice and Eve can take, we consider a large class of monitored quantum dynamics. These dynamics include Hamiltonian dynamics interspersed with measurements (Alice takes unitaries as the Trotterized evolution of a given Hamiltonian), or a given unraveling of a Lindblad modeling a generic open quantum system [47, 50, 81]. Thus, by identifying the coherence requirements for Alice to win the game, we identify the coherence requirements for communicating quantum information in a large class of dynamics.

We first approach this problem for the hybrid circuits generated by randomly chosen unitaries and measurements, and consider the dynamics resulting from Alice and Eve playing "random strategies" in which they pick the unitaries and measurements they perform from a random distribution. This is similar to approaches that use tools from random matrix theory to study entanglement dynamics and quantum chaos using random unitary circuits [31, 82–86]. Here, we also use randomness to derive general statements, and focus our investigation on coherence requirements. Then in section V A 1, we derive lower bounds for the coherence required for Alice to win regardless of the specific strategy she takes. In the main body of the paper, we restrict Alice and Eve to apply Clifford operations, to allow for numerical and analytic control over the problem. In section VI A, we discuss and speculate on the coherence resource requirements in games played with any choice of unitaries.

#### 1. Random hybrid circuit model

We first assume that Alice and Eve can only apply operations probabilistically, such that the information game can be considered as a random circuit composed of a sequence of $N$ steps, where at each step, $n$, one of the following operations are performed with a given probability:

- **CNOT on random site $i$ controlling a neighboring site $j = i \pm 1$ with probability $p_u$.** Throughout the paper we set $p_u = 1$ except in section IV A. Such gates keep constant the coherence in the $X$ and $Z$ bases, $C_x$ and $C_z$, because they reversibly map $Z$ basis states $|z_i, z_j\rangle$ to $Z$ basis states $|z_i, z_j \oplus z_i\rangle$, and $X$ basis states $|x_i, x_j\rangle$ to $X$ basis states $|x_i \oplus x_j, x_j\rangle$.

- **measurement of a random Pauli operator $A_i$ on a random site with probability $p_m$** With probabilities $p_m p_a$, the Pauli operator $A = (X, Y, Z)$ is chosen for $a = (x, y, z)$ respectively. The measurement destroys superposition in basis $A$ and reduces the coherence $C_a(|\psi\rangle)$. Since the probabilities $p_a$ control the relative rate of measurement, we fix $p_y = 1 - p_x - p_z$.

- **phase gate $R_z = \exp(i(Z_i + 1)\pi/4)$ with probability $p_R$** which can both increase and decrease $C_x$, but keeps constant $C_z$; and

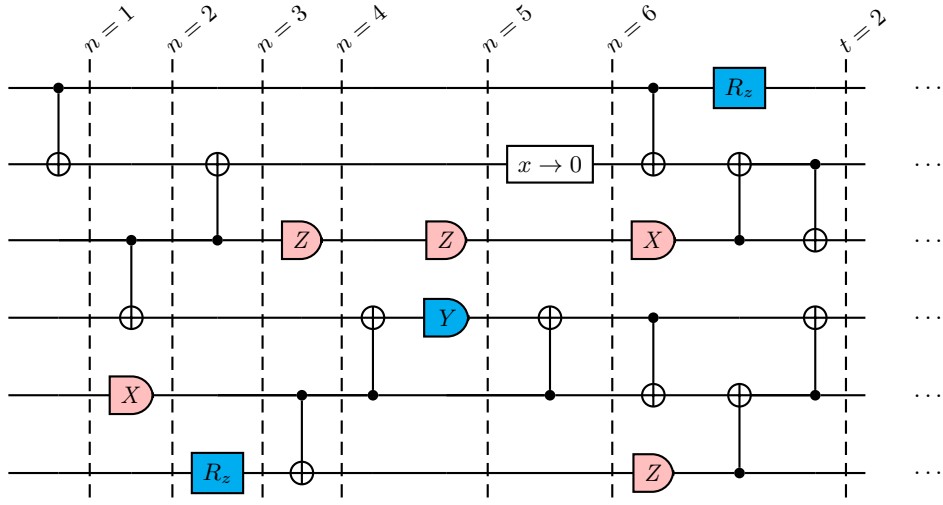

Figure 3. An example of the random hybrid circuit studied in the text. Each gate has an effect on the coherence in either the $X$ or $Z$ basis: preserves coherence in both basis (white), generates coherence in $X$ basis (blue), or destroys coherence in either $X$ or $Z$ basis (red). At each step $n$, a CNOT gate is applied to two neighboring qubits with probability $p_u = 1$, a measurement in the $X$, $Y$ and $Z$ basis is performed at a random site with probabilities $p_m p_x$, $p_m p_y$ and $p_m p_z$ respectively, a phase gate $R_z$ is applied with probability $p_R$, and a bit eraser error occurs with probability $p_e$. Time is measured as $t = n/L$ and since a CNOT is performed at each time step ($p_u = 1$), the circuit above shows $t = 2$ after $12 = 2L$ random CNOTs have occurred.

- **classical bit erasers** defined in section III B occurring with probability $p_e$.

After $n = L$ steps, $L$ CNOTs will have been applied ($p_u = 1$), so we measure time in units of $L$ steps, $t = n/L$. An example random circuit is shown in Fig. 3, and which operations are studied in which section is summarized in Table. II A 1. Since the CNOTs and $X$ measurements don't generate coherence in the $X$ basis, they are free operations for the coherence resource theory in that basis. While projective measurements of the $Y_i$ or $Z_i$ Pauli operator force the $i^{th}$ site to contribute 1 bit to the relative entropy of $X$ coherence, possibly increasing coherence, and are therefore not free operations of the $X$ coherence resource theory. While our approach focuses on the coherence in the $X$ basis, their exists a duality for CNOT gates which allows equal considerations for the coherence in the $Z$ basis: a CNOT gate on site $i$ controlling $j$ in the $Z$ basis is equivalent to a CNOT gate on site $j$ controlling site $i$ in the $X$ basis. This is particularly useful in section IV where we will discuss $C_x$ and leave implicit the dual result for $C_z$.

Finally, we assume that Alice knows the site and Pauli operator of all measurements performed, and records all their outcomes. In this way, she can keep track of the pure state which evolves in her qubits and potentially decode with a unitary operation at the end of the game. To identify her capability to decode, we will consider information quantifiers computed on the pure states and averaged over circuit realizations and possibly the measurement outcomes. For an experiment to observe the information quantifiers for one of the pure state produced (corresponding to a fixed set of measurements) they must repeat the experiment multiple times and wait

for the same fixed set of measurement outcomes to occur again. This procedure, called postselection, requires repeating the experiment a number of times exponentially large in the number of measurements performed and is a known [31, 34, 35] obstacle for observing measurement-induced phase transitions. For the purposes of this work, this obstacle is not particularly relevant, because 1) the circuits we consider can be simulated efficiently on a classical computer, and 2) the goal of our work is to identify the role of coherence in quantum communication as apposed to study quantum complexity.

## C. Stabilizer state tools

All gate operations discussed in this article, and presented in the section II B 1, are either part of the Clifford unitary group or are measurements of Pauli operators. This allows [72, 87] us to simulate the dynamics of these circuits efficiently using stabilizer states. Throughout, all numerical results presented are averaged over $O(2 \times 10^3)$ circuit realizations and all possible measurement outcomes for each circuit. The latter is possible because while different measurement outcomes, do result in different states, they do not result in different entanglement entropies for Clifford circuits. The stabilizer state tools also provides us with a strong analytic method to make predictions about the dynamics of these circuits. The details of these arguments are presented in the appendices.

### D. Unitary limit of coherence non-generating dynamics

If Alice is limited in her ability to produce superposition states, then she will generally be limited in what type of information she can encode. For example, even if Eve is not interfering with her qubits, $p_e = p_m = 0$, then Alice is still limited in the amount of entanglement she can generate if she is restricted to performing the free operations of either the $X$ and $Z$ coherence resource theories. This constraint, previously understood in a general setting [88–92], takes the following particularly useful form of the following theorem:

**Theorem 1.** *Given any local Pauli basis $D$, over $L$ qubits, and a pure state $|\psi\rangle$, the von Neumann entanglement entropy $S(\rho_r)$ for the reduced density matrix $\rho_r = \mathrm{Tr}_{A^c} [|\psi\rangle \langle\psi|]$ of any subsystem $R$ is bounded by the coherence of the local Pauli basis:*

$$S(\rho_r) \leq C(|\psi\rangle, D). \tag{2}$$

Here a Pauli basis $D$ is any basis diagonal in a set of chosen Pauli operators $\{A_i\}$ with $i \in (1 \dots L)$ and $A_i \in (X_i, Y_i, Z_i)$.

*Proof* The proof is given by the set of inequalities $C(|\psi\rangle, D) = H(P(d)) \geq H(P_r(d_r)) \geq S(\rho_r)$ where $P_r(d_r)$ is the marginal distribution of the $P(d)$ defined on the subsystem $R$, $S(\rho)$ is the von Neumann entropy and $H(P)$ is the Shannon entropy. The first inequality is because the Shannon entropy of a bipartite distribution is greater than any of its marginals, and the second inequality follows from the data processing inequality for the von Neumann entropy [93], which states the von Neumann entropy is constant or increasing under any CPTP map. Here the CPTP map is taking the diagonals of $\rho_r$ in the Pauli basis $d_r$: $\rho_r \to \sum_{d_r} |d_r\rangle \langle d_r| \langle d_r| \rho_r |d_r\rangle = \rho_{d_r}$. The second inequality then follows from $H(P_r(d_r)) = S(\rho_{d_r}) \geq S(\rho_r)$. □

Therefore, if Alice is only able to apply CNOTs (free operations in both the $X$ and $Z$ coherence resource theory), then she will not be able to increase the coherences $C_x$ or $C_z$, and the entanglement of the states she can produce will be limited accordingly (i.e $S(\rho_r) \leq \min(C_x, C_z)$). This constraint is explicitly revealed in the steady state entanglement of the random circuit containing only CNOTs ($p_m = p_e = p_R = 0$). In this circuit, the coherences $C_x$ and $C_z$ are conserved quantities since the CNOTs neither increase nor decrease the coherence in the $X$ or $Z$ basis. In conjunction with the above theorem, the conservation of $C_x$ and $C_z$ implies that the von Neumann entropy of any subsystem is bound by the coherence $C_x$ and $C_z$ in the initial state. If we consider an initial product state with $N_x$ qubits polarized in the $X$ direction and $N_z = L - N_x$ qubits polarized in the $Z$ direction, we find that the coherences at all times is $C_x = N_z$ and $C_z = N_x$. At late times, the CNOTs gates drive the system through an exponentially large in $L$ number of states, most of which are maximally entangled subject to the bound in Theorem 1. In addition

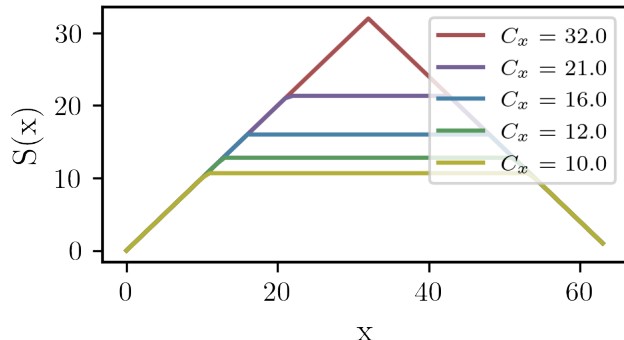

Figure 4. Late time entanglement distribution for the random CNOT circuit ($p_m = 0$, $p_R = 0$ and $p_e = 0$). For such a CNOT circuit, the coherence $C_x$ is conserved, and the coherence of the initial state is shown in the legend. At late times, the above entanglement distribution can be predicted under the assumption that the entanglement in a region is maximized subject to the constraint given by Theorem 1.

to this bound, the entanglement entropy of any given subregion $A$ of the system is limited by the size of that subregion: $S(\rho_r) \leq |A|$. Combining these two bounds, we predict that the entanglement entropy of a region between sites 1 and $x$ is

$$S(x) = \min(x, L - x, C_x, C_z). \tag{3}$$

This is confirmed in Fig. 4, where we show that the late time entanglement entropy, $S(x)$ for $L = 64$ site system with $C_x \leq C_z$ and demonstrate that the bound $C_x \geq S(\rho_r)$ is saturated for any subsystem with size $|A| \geq C_x$. In that figure, and in numerical results presented throughout the manuscript, statistical fluctuations due to circuit sampling are suppressed in system size. This is a generic feature of entanglement growth [50, 83] in random circuit models.

### III. COMMUNICATION IN COHERENCE LIMITED RANDOM CIRCUITS

#### A. Alice protects a classical diary

We begin our investigation by considering what type of information Alice can protect if she only has access to $X$ coherence-free states, with $C_x = 0$, and is only able to apply Incoherent Operations in the $X$ basis. From Theorem 1, she is unable to produce entangled states from pure states (and more generally from mixed [92]), and is, therefore, unable to encode information non-locally. Furthermore, she cannot even create superposition in the $X$ basis and, therefore, cannot encode quantum information locally. In this section, we will show that while she cannot protect quantum information, she is able to encode and protect classical information given that Eve is limited in the rate at which she can induce errors in Alice's

qubits.

First imagine that Alice prepares an X-basis state, $|x\rangle$, encoding some classical information in a classical bit string $x = (x_1, x_2, \dots x_L)$ with bits $x_i \in (0, 1)$. Eve then begins an attack by applying bit erasers at random sites, such that Alice's $i^{th}$ bit evolves as $x_i \to 0$ when the bit eraser is applied there. In this section, we model this attack using a local quantum channel with Kraus operators $E_{1,i} = |0\rangle_i \langle 0|_i$ and $E_{2,i} = |0\rangle_i \langle 1|_i$, where $|0\rangle_i$ and $|1\rangle_i$ are the eigenstates of $X_i$. In the next section we describe how this channel can be implemented using measurements. Since this channel can only destroy coherence, it is a free operation of the $X$ coherence resource theory, and the coherence in the $X$ basis will remain 0 after the random sequence of CNOTs and bit erasers. Furthermore, these operations map $X$ basis state to $X$ basis state, and the evolution of the qubits is described by a sequence of classical maps between $X$ bit strings $x_n \to x_{n+1}$. Accordingly, these dynamics can be considered a classical limit of the hybrid quantum circuits previously discussed.

We now consider the dynamics when random strategies are played, and identify the maximum rate at which Eve can create errors such that Alice is able to protect her classical diary by applying the sequence of random CNOTs. For a specific choice of strategies, a classical map $x_0 \to x_n = f_n(x_0)$ on the bit strings $x_0$ describes how an initial basis state $|x_0\rangle$ maps to the basis state $|x_n\rangle$, after $n$ steps of the game. To estimate the average number of bits Alice can store and recover, we assume the initial bit string is sampled from a uniform distribution $P_0(x) = 2^{-L}$ and evaluate the mutual information

$$I_x = H(P_n) + H(P_0) - H(P_{0,n}), \qquad (4)$$

where, $P_{0,n}(x_0, x_n) = P_0(x_0)\delta(x_n - f_n(x_0))$ is the joint distribution between initial, $x_0$ and final bit strings $x_n$, $P_n(x_n) = \sum_{x_0} P(x_0, x_n)$ is the distribution of final bit strings, and $H(P)$ is the Shannon entropy of the distribution $P$. Via the noisy-channel coding theorem [94], this gives a lower bound on the number of bits Alice can store, given the proper encoding and decoding scheme.

In Fig. 5 we show the mutual information $I_x$ averaged over 1000 games and demonstrate that Alice can indeed protect classical information at long times, so long as the rate at which Eve can attack her qubits is limited (i.e. $p_e < 0.1$). At $p_e \approx 0.1$, the circuit undergoes a phase transition from a finite channel capacity to vanishing channel capacity. For $p_e > 0.1$, the mutual information vanishes and Alice is unable to recover her diary from the late time state $x_n$.

### 1. Purification transition for classical circuits

This result was obtained numerically by studying the purification of an initially mixed state similar to results for quantum channel capacity [26]. A similar approach

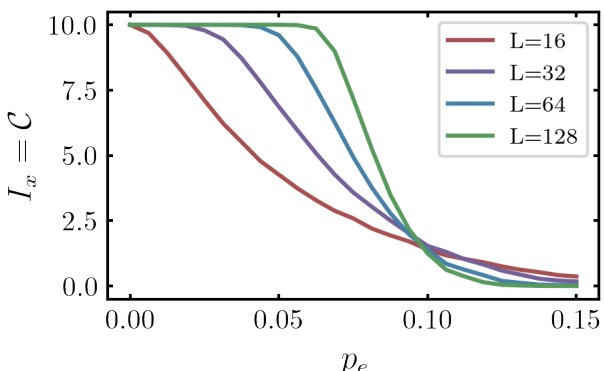

Figure 5. Classical, $I_x$, and quantum, $\mathcal{C}$, channel capacities of a random hybrid circuit composed of CNOTs and coherence-maintaining bit erasers as a function of $p_e$ for different lengths $L$ shown in the legend. The random hybrid circuit is composed of first a set of $t = 40L$ CNOT gates followed by the hybrid circuit ($p_m = p_R = 0$) for a time $t = 10L$. The channel capacities were computed via purification dynamics with $A = 10$ ancillas. Depending on if the initial state has classical or quantum correlations, the dynamics of the purity determine the classical channel capacity $I_x$ or the quantum channel capacity $\mathcal{C}$. As discussed in the text, they are the same, $I_x = \mathcal{C}$, when we consider a channel composed of CNOTs and coherence-maintaining bit erasers.

was taken in Refs. [26–30]. There, the measurement-induced phase transition in the quantum channel capacity of a given circuit was equated to the purification dynamics of an initial mixed state evolved under that same circuit. In the coherence-free circuits, a similar relation also holds for the classical channel capacity. We derive this relation by introducing a set of ancilla bits, $A$, used to store $|A|$ classical bits of the initial state of the system and which do not undergo any dynamics as the system, $S$, evolves under the random hybrid circuit. For studying the classical channel capacity this classical memory is achieved by initializing the system in the correlated state:

$$\rho_{SA}(n = 0) = \frac{1}{2^A} \sum_{x \in S_a} |x, x, 0\rangle \langle x, x, 0| \qquad (5)$$

where $S_a$ is all $2^A$ bit strings of length $|A|$, and $|a, s_1, s_2\rangle$ (equivalently $\langle a, s_1, s_2|$) is an $X$ basis state of the system and ancilla with the ancilla bit string, $a$, the first $|A|$ bits of the system $s_1$, and the last $L-|A|$ bits of the system $s_2$. Such a mixed state has system $S$, ancilla $A$ and the joint system and ancilla $A \cup S$ in a mixed state with von Neumann entropy $S(\rho_S) = S(\rho_A) = S(\rho_{AS}) = |A|$. Since the density matrix is diagonal in the $X$ basis, the Shannon entropy for the $X$ basis states is equivalent to the von Neumann entropy, and the mutual information between system and ancilla is therefore $I_x = |A|$, reflecting the fact the ancilla remembers perfectly the initial state of the system. After evolution of the random hybrid circuit,

the system-ancilla state evolves to

$$\rho_{SA}(n) = \frac{1}{2^A} \sum_{x \in S_a} |x, f_n((x,0))\rangle \langle x, f_n((x,0))| \quad (6)$$

where $|x, f_n((x,0))\rangle$ is an $x$ basis state with the ancilla in state $a = x$ and the system is in state $(s_1, s_1) = f_n((x,0))$. Such a state still has ancilla and joint entropy $S(\rho_A) = S(\rho_{SA}) = |A|$, but with classical mutual information $I_x = S(\rho_S)$ that depends on the channel capacity of the classical evolution $f_n((x,0))$. If the channel capacity (classical mutual information) goes to 0, then the entropy of the system $S(\rho_S) = I_x$ also goes to 0 and the system purifies. While instead, in the error protecting phase, the system remains mixed with $S(\rho_S) = I_x > 0$. The transition between the two phases has been argued to be within the directed percolation universality class [75, 76].

## B. Alice protects a quantum diary against coherence preserving errors

While in the last section, we concluded that Alice can not encode quantum information without being able to generate $X$ coherence, we can still ask if she can protect a state that already is encoding quantum information. Specifically, we now investigate the restrictions that must be placed on Eve's bit eraser procedure such that Alice, given a quantum state with coherence $C_x > 0$, can protect quantum information encoded in that state. While above we determined that the random CNOTs allow memory of an initial $X$ basis state, we are now interested if they can also remember an arbitrary superposition of a set of the $X$ basis states at late times. As we will see below, this extra requirement translates to an extra requirement on how the bit erasers are implemented.

If we take the quantum bit eraser with Kraus operators $E_{1,i}$ and $E_{2,i}$, then they will destroy coherence and quickly erase any superposition in the initial state. This is seen by the following measurement implementation of these Kraus operators:

1. measure $X_i$;

2. flip $X_i$ if $x_i = 1$;

3. forget measurement outcome $x_i$ (don't perform postselection);

which acts on the state $|\psi_c\rangle = \psi_1 |\phi_1, 1\rangle + \psi_0 |\phi_0, 0\rangle$ as:

$$|\psi_c\rangle \langle \psi_c| \rightarrow |\psi_1|^2 |\phi_1, 0\rangle \langle \phi_1, 0| + |\psi_0|^2 |\phi_0, 0\rangle \langle \phi_0, 0| \quad (7)$$

Notice, that while forgetting the measurement outcome is required to implement the Kraus operators, the coherence is still destroyed if the measurement outcome is recorded and postselected. We will refer to the channel without postselection as the *quantum bit eraser*, and the channel with postselection as the *coherence-destroying bit eraser*.

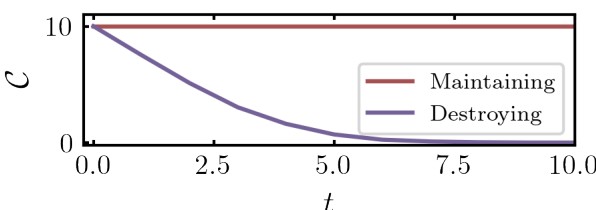

Figure 6. Difference in quantum channel capacity $\mathcal{C}$ between the coherence-maintaining bit eraser and coherence-destroying bit eraser for a system of $L = 128$ bits for $p_e = 0.02$. After $t = |A| = 10$ coherence-destroying bit erasers are applied, the coherence and quantum channel capacity approaches zero. For $p_e = 0.02$ this occurs at times $t = 10/p_e/L \approx 4$ as shown in the plot.

This is in contrast to the *coherence-maintaining bit eraser* which is implemented by the following sequence of measurements and unitaries:

1. measure $Z_i$;

2. post select the outcome $z_i = 1$;

3. measure $X_i$;

4. flip $X_i$ if measurement outcome was $x_i = 1$;

which has an action on the state $|\psi_c\rangle$ as:

$$|\psi_c\rangle \rightarrow (\psi_1 |\phi_1\rangle + \psi_0 |\phi_0\rangle) \otimes |0\rangle \quad (8)$$

and preserves coherence so long as $\langle \phi_1 | \phi_0 \rangle = 0$. For example, take $|\psi_c\rangle$ as the Bell state $|\psi_c\rangle = (|00\rangle + |11\rangle)/\sqrt{2}$ which has one bit of coherence in the $X$ basis when $|00\rangle$ and $|11\rangle$ are $X$ basis states. After performing the coherence-maintaining bit eraser on the first site, the state $|\psi_c\rangle$ becomes $(|00\rangle + |01\rangle)/\sqrt{2}$ which still has $C_x = 1$. This is contrast to the coherence-destroying bit eraser, for which the $X_1$ measurements learns the $X_2$ state of the first qubit and reduces the coherence to $C_x = 0$.

The difference is revealed by considering the quantum purification transition which is directly related to a transition in the quantum channel capacity, and will allow us to identify which bit eraser efficiently destroys quantum information. In the quantum purification transition, instead of initializing the system and ancilla with the classically correlated mixed state in Eq. 5, the system is initialized in the entangled state:

$$|\psi_{SA}\rangle (n = 0) = \frac{1}{2^{|A|}} \sum_{x \in S_a} |a = x, s_1 = x, s_2 = 0\rangle \quad (9)$$

such that the ancilla is now remembering the initial quantum state of the first $|A|$ system bits. Note that as in the classical case, the purpose of the ancilla is only to encode the initial state and accordingly, no gates or measurements are applied to it as the hybrid circuit evoles.

While the system reduced density matrix still has no coherence, the system and ancilla together have coherence $C_x = |A|$ reflecting the encoding of quantum information; this is contrast to the state encoding of classical information in Eq. 5, which has $C_x = 0$. The quantum channel capacity is then quantified by the coherent information [26], $\mathcal{C} = S(\rho_S) - S(\rho_{SA})$, and so long as the system-ancilla remains pure (as is the case for the coherence preserving and destroying bit erasers) then $S(\rho_{SA}) = 0$ at all time such that the coherent information is $\mathcal{C} = S(\rho_S) = S(\rho_A)$. Such a quantum channel capacity gives an upper bound on the number of qubits of the initial quantum state recoverable from the final quantum state of the system [95–97].

For the coherence preserving bit eraser, the dynamics of the system reduced density matrix is equivalent to that discussed in section III A and thus give $S(\rho_S) = I_x$ as before, but now $S(\rho_{SA}) = 0$ such that $\mathcal{C} = S(\rho_S) = I_x$. Thus, for the coherence preserving bit erasers, the system protects quantum information as long as it protects classical information. In contrast, for the coherence destroying bit eraser, the state $\rho_{SA}$ loses $X$ coherence after $O(|A|)$ bit erasers, and becomes an $X$ basis state with zero entanglement between system and ancilla. Thus, the system and ancilla purify, coherent information is lost $\mathcal{C} \to 0$, and Alice is unable to recover her quantum diary. This distinction in the dynamics of the coherent information between the two types of errors is shown in Fig. 6.

## C. Dynamically evolving classical codes

Before continuing our discussion on the information game played between Alice and Eve, we will first discuss how the above results in section III A and section III B are directly related to error correction codes. We will first make the connection explicit by showing the dynamics above can be interpreted as the dynamics of a classical code space and by describing how this code space evolves. Then, we will argue that Alice's classical diary is encoded in this space, while her quantum diary would be a superposition over states in this space similar to the design of the repetition code. This connection between the coherence-free hybrid circuits and repetition codes, which can only correct bit errors, hints at the first connection between coherence and the type of errors a code can correct. We will then continue discussing the information game in section IV where we allow Eve to apply coherence generating measurements.

The connection between error correction codes and the dynamics in the above section is because the purification dynamics of the mixed state on Alice's system $\rho_S = \mathrm{Tr}_A \rho_{SA} = \mathrm{Tr}_A |\psi_{SA}\rangle\langle\psi_{SA}|$ map directly to the evolution of a classical linear code space. In particular, we show that the reduced state of the system, $\rho_S(n)$, discussed in section III A and section III B, can be written in the following form (See Appendix A 4 for proof):

$$\rho_S(n) = \frac{1}{2^{k_n}} \sum_x |x\rangle\langle x| \prod_{i=1}^{L-k_n} \delta\left(\sum_j H_{ij}^x(n)x_j\right)$$

where $H_{ij}^x(n)$ is an $L - k_n$ by $L$ matrix describing the allowed $x$ basis states in $\rho_S(n)$, and $k_n = S(\rho_S(n))$. This expression shows that only basis states that satisfy the constraint $\sum_j H_{ij}^x(n)x_j = 0$ are allowed to occur in the evolving mixed state, and that these basis states all have equal probability of occurring. Since this constraint has the same form as the parity check matrix determining the code words in a classical linear code space [72], we can consider the dynamics of the state $\rho_S(n)$ as the dynamics of a classical linear code space, $\mathcal{K}^x(n)$, composed of those code words. Remember here that a code space is defined as the set of bit strings, $\mathcal{K}^x(n)$, that can be used to encode a message, and that the code space can be defined by a check matrix, $H^x$, as the set of bit strings satisfying the above constraint: $x \in \mathcal{K}^x(n)$ if and only if $\sum_j H_{ij}^x(n)x_j = 0$ for all $i$.

When a random sequence of CNOTs is applied to the state, the different code words evolve under the same random sequence of CNOT gates, and the difference between the code words, measured by the Hamming distance $d(x_1, x_2) = \sum_i |x_{1,i} - x_{2,i}|$ generically increases. On the other hand, a bit eraser on the $i^{th}$ bit will decrease the Hamming distance between two code words with $x_{1,i} \neq x_{2,i}$. If there are too many bit erasers, the Hamming distance between two different code words will shrink to zero, and those two code words will become the same bit string. When this occurs, the number of distinct code words, $S(\rho_S)$, will decrease resulting in the system purifying and a loss of channel capacity $I_x = S(\rho_S)$. Thus, the loss of channel capacity is equivalent to the shrinking of the evolving classical code space.

Finally, by considering the full ancilla and system state, we can see that Alice's diary is encoded in this classical code space. That is, any $k$ bit classical message she wishes to encode can be represented in her system by one of the classical code words in the evolving code space. By considering the system and ancilla state in Eq. 6, we find that the initial basis state of the system $|(x, 0)\rangle$ is mapped to the basis state $|f_n((x, 0))\rangle$ at a later time. Upon tracing out the ancilla, we find $|f_n((x, 0))\rangle$ must be an allowed basis state and lives in the classical code space $\mathcal{K}^x(n)$. Thus, if Alice encodes a $k$ bit message in initial state $|(x, 0)\rangle$, then that message will become encoded in the evolving code space $\mathcal{K}^x(n)$. Similarly, Alice's quantum diary would be encoded on some superposition of code words in the evolving code space.

This is similar to how a quantum bit in the repetition code is encoded in the classical code space with code words $x_1 = (111\dots)$ and $x_0 = (000\dots)$. Notice that in both the information game and in the repetition code, bit flip errors can be corrected but only if they don't include a phase or decoherence error. In the information game, we showed the issue was Alice's inability to maintain and

regenerate coherence, and by analogy we should expect coherence can also provide a more general context for why the repetition code can't correct phase errors. Below, in section V B 1, we show that this more general context is the Theorem 3 which gives how coherence bounds the code distance of a quantum error correction code.

## IV. MEASUREMENT INDUCED DYNAMICS OF COHERENCE

In the previous section, both Alice and Eve were restricted to applying Incoherent Operations in the $X$ basis, and depending on whether Eve performed a coherence-destroying or coherence-maintaining bit eraser, Alice was able to protect a quantum diary. In this section, we will allow Eve to make $Y$ and $Z$ measurements which are not Incoherent Operations in the $X$ basis and can potentially increase the coherence in Alice's system. While this would not be an optimum strategy for Eve, she may not have control over which basis she measures in and we can therefore investigate if Alice can take advantage of coherence generating measurements. Below, we find that this is the case, and show that Alice can protect a quantum diary when Eve performs $Y$ measurements at a sufficiently high rate $p_y > |p_x - p_z|$.

We start by investigating how the coherence of a pure state evolves under such a random hybrid circuit generated by Alice and Eve playing random strategies. For Alice to be able to take advantage of the coherence generated by the measurements, then the steady state coherence in the $X$ and $Z$ basis, $C_x$ and $C_z$, must scale with system size. Otherwise, the entanglement will be constrained, via Eq. 3, to be sub-extensive, Alice will only be able to encode information locally, and her diary will be susceptible to local errors.

### A. Measurement only limit

We begin by considering the dynamics in the measurement-only limit ($p_u = p_R = p_e = 0$ and $p_m = 1$) for an initial product state with $N_x$, $N_z$ and $N_y = L - N_x - N_z$ qubits polarized in either the $X$, $Z$ or $Y$ directions respectively. In this limit, the evolving state remains a product state, but with a different number of qubits polarized in a given direction. States of this form have $N_y + N_z$ qubits uncertain in the $X$ direction and therefore have $C_x = N_y + N_z = L - N_x$ qubits of $X$ coherence. Similarly for the coherences in the $Z$ and $Y$ directions: $C_z = L - N_z$ and $C_y = L - N_y = L - C_x - C_z$.

The coherences then evolve according to how the number of qubits polarized in a given direction is randomly updated after a given measurement. At each step, $n$, a random measurement is made and the number of qubits polarized in a given direction, $N_\alpha$, can change by at most one. Whether they change or not depends on the type of measurement made and the probability that measure-

ment is made on a site polarized in a direction different from the measurement basis. That probability depends only on the value of $N_\alpha$ before the measurement and so the stochastic process for which $N_\alpha$ are updated is Markovian. The conditional probabilities of this process are derived in Appendix D 1 and lead to the following rate equation:

$$\partial_m \overline{N}_x(m) = p_x \frac{L - \overline{N}_x}{L} - (p_z + p_y) \frac{\overline{N}_x}{L}, \qquad (10)$$

with similar equations for $\overline{N}_y$ and $\overline{N}_z$, and where the overlie in $\overline{N}_\alpha$ refers to averaging over circuit realizations. Importantly, the rate at which $\overline{N}_x$ increases or decreases depends on the number of qubits already polarized in the $X$ direction. This follows from the fact that the effect of a measurement depends on the polarization of the bit it is applied to: $X$ measurements only increase $N_x$ if they are applied to a $Y$ or $Z$ polarized bit. The steady state solution to these dynamics predicts that the average steady state density of $\alpha$ polarized qubits is equal to $p_\alpha$ as intuitively expected: $\overline{N}_\alpha = p_\alpha L$ or equivalently for the coherences $C_\alpha = (1 - p_\alpha) L$. Thus, if the dynamics of coherence in the measurement-only limit were robust to the addition of unitary gates, Alice may be able to make use of the volume-law coherence to encode states non-locally.

### B. Random walk of coherence in weak measurement limit

Unfortunately for Alice, this is not the case as shown by studying the weak measurement limit where the measurement rate $p_m$ is so small that the system maximizes entanglement with respect to the bounds given by coherence (c.f. Eq. 3). In this limit, the coherence dynamics again becomes Markovian because, as we show in Appendix D 2, every measurement changes the coherence in the $X$, $Y$ or $Z$ basis independently from the current coherence in the system. Roughly, this can be expected because any time a measurement occurs in this limit, each qubit will be in a maximally mixed state (as long as $\min(C_x, C_z) > 1$) and a measurement on any qubit is guaranteed to have an uncertain outcome and change the state and coherence. This is in contrast to the measurement-only limit where the effect of a measurement strongly depends on the current coherences in the system. This limit occurs when, between measurements, there are enough CNOTs performed to guarantee that a sequence of CNOTs can entangle any two distinct qubits within the system. This will occur after $n = O(L^2)$ CNOT gates and thus we require that $p_m < 1/L^2$.

In Appendix D 2, we derive the Markov stochastic process for $N_x$ and $N_z$, where instead of being the number of qubits polarized in the $X$ or $Z$ direction, $N_x = L - C_x$ and $N_z = L - C_z$ are now, more generally, the number

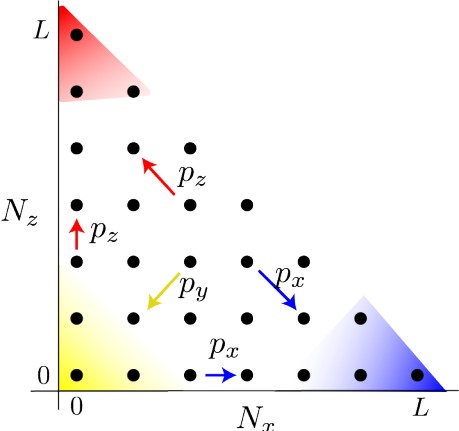

Figure 7. Cartoon of the random walk in the information known about the $X$ and $Z$ basis: $N_x = L - C_x$ and $N_z = L - C_z$ respectively. A measurement in the $X, Y$ or $Z$ direction occurs with probability $p_x$, $p_y$, and $p_z$, creating jumps in information known about each basis $N_x$ and $N_z$ as shown. The walker can not go above the line $N_x + N_z = L$ because the incompatibility between the $X$ and $Z$ observables restrict the total amount of information known about the $X$ and $Z$ bases. The walker localizes in one of the corners of the plane depending on which which measurement rate dominates (shown in red, yellow and blue). When $Y$ measurements dominate, information about the $X$ basis and $Z$ basis is lost and the walker localizes in the $(N_x, N_z) = (0, 0)$ corner. While when $Z$ or $X$ measurements dominate, information about the $X$ or $Z$ basis becomes maximum and the walker localizes in the respective corners.

of bits of information that can be specified about the $X$ or $Z$ basis states. Where as $C_{x(z)}$ gives the entropy, or the number of uncertain bits, of a given state's distribution in the over the $X(Z)$ basis, $N_x = L - C_x$ give the lack of entropy, or the number of bits known about the basis states. We find that the stochastic process is a biased random walk in the $(N_x, N_z)$ plane, subject to the bounds $0 \leq N_{x(z)} \leq L$ and $N_x + N_z \leq L$ and is described in Fig. 7. The direction of the drift velocity of the walker is determined by the relative rates of the measurements and yields the rate equations:

$$\partial_m \overline{N}_x(m) \sim p_x - p_z - p_y, \qquad (11)$$
$$\partial_m \overline{N}_z(m) \sim p_z - p_x - p_y.$$

These rate equations predict that when $p_x > p_z + p_y$, the probability distribution for the walker localizes around the point $(N_x, N_z) = (L, 0)$ with a localization length proportional to $\lambda \sim 1/(p_x - p_z - p_y)$. Since the rates of the Markov process are constants, the localization length does not depend on system size such that $\overline{N}_x = L - c\lambda$ and so the average coherence $\overline{C}_x = c\lambda$ becomes an area-law, where the constant $c$ depends on the detailed features of the walker distribution. In this limit the evolving quantum state is mostly classical in the $X$ basis and, by Theorem. 1, can only support area-law scaling of entanglement. Thus, Alice will not be able

to encode quantum information non-locally and her diary will be susceptible to local errors. Similarly, when $p_z > p_x + p_y$, the walker localizes around the point $(N_x, N_z) = (0, L)$; the coherence $C_z \to 0$; states become classical in the $Z$ basis; and volume-law states are again forbidden. Again, Alice will not be able to protect a quantum diary.

In the region $p_x > p_z + p_y$, the walker becomes less localized as $p_x$ is decreased, until the point $p_x = p_z + p_y$ at which the localization length diverges and the walker distribution becomes uniformly distributed along the $N_z = 0$ axis. At this critical point of the random walk, the coherence in the $Z$ direction remains maximal $\overline{C}_z \sim L$, while the average coherence in the $X$ direction becomes $\overline{C}_x = \overline{N}_x = L/2$ giving rise to the possibility of volume states and the ability of Alice to protect quantum information. Decreasing $p_x$ further, the walker becomes localized around the point $(N_x, N_z) = (0, 0)$ where both coherences are scaling with volume and volume-law entangled states will be allowed. In this limit, $p_y > |p_x - p_z|$, Alice has access to volume-law coherence and can potentially use it to protect a quantum diary.

These three limits are summarized in Fig. 2, where we describe the regions of the $(p_y, \Delta_x = (p_x - p_z)/(1 - p_y))$ plane where $C_x \to 0$ (X-classical states appear), $C_z \to 0$ (Z-classical states appear) and where both coherences scale with the volume of the system (region labeled "Quantum"). In this "Quantum" region, volume-law entangled states are possible and, as we show in the next section, Alice is able to protect quantum information.

## C. Entanglement criticality at finite measurement rate

While the above discussion relies on assumptions valid only when $p_m < 1/L^2$, it appears to qualitatively capture numerical simulations for finite $p_m = 0.01$. In Fig. 8, we plot the $X$ coherence, $C_x$, and half cut entanglement entropy, $S(L/2)$, in the $(p_y, \Delta_x)$ plane and find both have a sharp change at $p_x = p_y + p_z$. When $p_y > |p_x - p_z|$, we observe both $C_x > L/2$ and $C_z > L/2$, while instead when $p_y < |p_x - p_z|$, one of the coherences drops below $L/2$ consistent with the above predictions for $p_m < 1/L^2$. This sharp change in coherence is accompanied by a transition from volume-law entanglement, $S(L/2) \sim L$, in the region $p_y > |p_x - p_z|$ to area-law entanglement in the region $p_y < |p_x - p_z|$. Furthermore, we observe in Fig. 10, and discuss further in section IV D, that in the quantum phase, $p_y > |p_x - p_z|$, Alice is able to protect a quantum diary with a number of qubits scaling with system size. The main difference from the weak measurement limit, where $p_m$ vanishes in the infinite size limit (i.e. $p_m \to 1/L^2$), is that the coherences remain volume-law throughout the phase diagram. We explain this discrepancy by deriving, in appendix D 3, a phenomenological rate equation for the finite measurement rate $p_m$ that interpolates between the measurement-only limit and van-

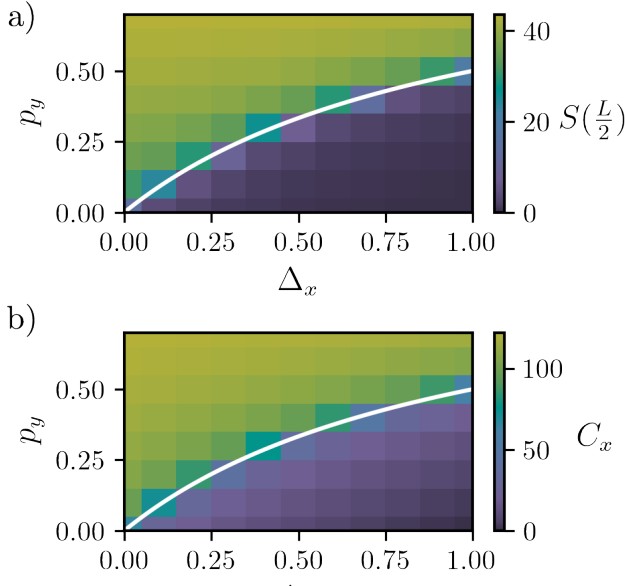

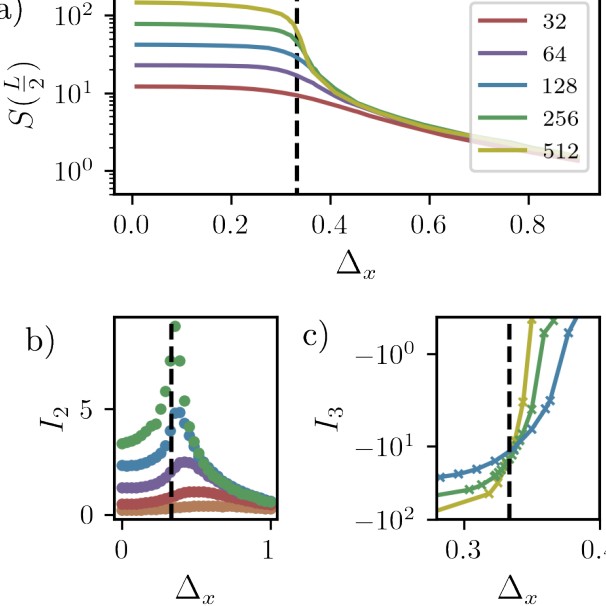

Figure 8. (a) Steady state half cut entanglement, $S(x = L/2)$, and (b) coherence, $C_x$, for the hybrid circuits with a sequence of random CNOTs and $X$, $Y$ and $Z$ measurements for the small measurement rate of $p_m = 0.01$ ($p_R = 0$ and $p_e = 0$). In these figures $\Delta_x$ parameterizes the imbalance between $X$ and $Z$ measurements $\Delta_x = (p_x - p_z)/(1 - p_y)$. These figures show the critical line (white), $|\Delta_x| = p_y/(1 - p_y)$, which is predicted by the competition between coherence loss and growth rates as discussed in the text. Due to the duality between the $X$ and $Z$ bases, dynamics are the same for $\Delta_x \to -\Delta_x$. For the $\Delta_x > 0$ side of the duality, the coherence in the $Z$ basis, $C_z$, is greater than $C_x$ and does not constrain the entanglement. These figures were computed with $L = 128$ and averaging over 2000 circuit realizations.

Figure 9. a) Half cut entanglement entropy $S(L/2)$ v.s. imbalance between $X$ and $Z$ measurements $\Delta_x = (p_x - p_z)/(1 - p_y)$. The $\Delta_x = 1/3$ critical point predicted by $p_y = |p_x - p_z|$ is confirmed by the b) Anti-podal mutual information $I_2$ and b) triparitite mutual info $I_3$ as discussed in the text. Different lines correspond to the system sizes shown in the legend. In this figure $p_m = 0.01$, $p_y = 1/4$ and $p_R = p_e = 0$.

ishing measurement limit.

This rate equation is constructed by introducing a length scale $\xi$, associated to the typical distance at which two qubits might be entangled. More precisely, it is the typical length [30] of a stabilizer (see appendix D 3), and scales with the half cut entanglement entropy $S(L/2) \sim \xi$. Under this assumption, we derive a rate equation for the dynamics of the coherence:

$$\partial_m \overline{N}_x = p_x \left(1 - \left(\frac{\overline{N}_x}{L}\right)^\xi\right) - p_z \left(1 - \left(\frac{\overline{N}_z}{L}\right)^\xi\right) \quad (12)$$
$$- p_y \left(1 - \left(\frac{L - \overline{N}_x + \overline{N}_z}{L}\right)^\xi\right).$$

In a volume-law phase, this length diverges with system size, $S(L/2) \sim \xi \sim L$, such that the phenomenological rate equation is equivalent to Eq. 11 as $L \to \infty$. If $p_y > |p_x - p_z|$ then the steady state of this rate equation is consistent with the assumption of a volume-law phase. Instead, if $p_y < |p_x - p_z|$, then Eq. 11 predicts that the coherence and entanglement entropy becomes area-law,

such that $\xi \sim S(L/2)$ must scale as constant with system size. Thus, when $p_y < |p_x - p_z|$ only area-law entanglement is consistent, and the predictions for the entangling phases at $p_m \to 1/L^2$ hold at finite $p_m$. The main difference is that $\xi$ is now finite when $p_y < |p_x - p_z|$, such that the rate Eq. 12 predicts volume-law coherence consistent with numerical simulation.

In Fig. 9, we confirm precisely the prediction for the critical point $p_y = |p_x - p_z|$ when measurements in the $Y$ direction are fixed at a rate $p_y = 1/4$. There, we observe a transition between area-law and volume-law entanglement at the critical point $|\Delta_x| = |p_x - p_z|/(1-p_y) = 1/3$ as predicted. Near the critical point, $\Delta_x \approx 1/3$, we find entanglement scaling logarithmically with system size which creates finite sizes obstacles [98] to an accurate determination of the critical point. Accordingly, we consider the anti-podal mutual information, $I_2$, and the tripartite mutual information, $I_3$, to obtain circumvent these finite size effects as done in Refs. [26, 30]. The anti-podal mutual information is computed as $I_2 = S(\rho_{R_1}) + S(\rho_{R_3}) - S(\rho_{R_1 \cup R_3})$, where the regions $R_n$ contain the sites from $(n-1)L/4$ to $nL/4$. This quantity is a constant when the entropies follow either volume or area-law, but scales logarithmically with $L$ if the entropies of the different subregions scale logarithmically. Thus, it is highly sensitive to the logarithmic scaling of entanglement at the critical point, and a sharp peak identifies the

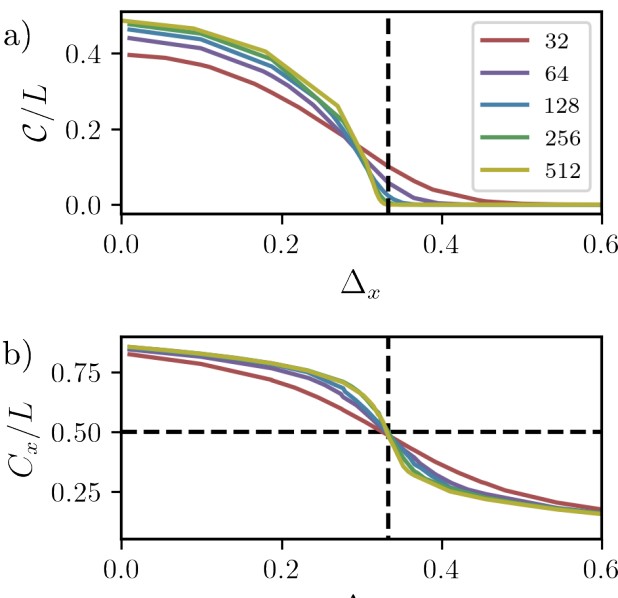

Figure 10. a) This figure shows the quantum channel capacity $\mathcal{C}$, computed by the purification dynamics, at a time $t = 5L$, for the same circuits studied in Fig. 9. It shows a transition at $\Delta_x \approx 1/3$ (marked by a black vertical dashed line), the location of the critical point determined by the competition of coherence generating and coherence destroying measurements $p_y > |p_x - p_z|$. b) Coherence for the same circuit but with an initial pure state. The coherence crosses $C_x = L/2$ at the critical point $\Delta_x = 1/3$ as predicted by the rate Eq. 12.

$\Delta_x = 1/3$ critical point in Fig. 9. Interestingly, it also suggests the volume-law phase has a logarithmic correction to the entanglement scaling; we leave for future work the task of identifying the origins of this correction.

The tripartite mutual information also shows the $\Delta_x = 1/3$ critical point and is computed as $I_3 = 4S(R_1) - 2S(R_1 \bigcup R_2) - S(R_1 \bigcup R_3)$, which is equivalent to the form in Ref. [26] due to translational invariance. The triparitite mutual information goes to 0 in the area-law phase, follows a volume-law in the volume-law phase, and it goes to a constant independent of system size at the critical point. Fig. 9 shows this behaviour with different curves for different system sizes $L$ crossing at the critical point, $\Delta_x = 1/3$.

### D. Transition in channel capacity

We have shown that when Eve applies a sufficiently high rate of $Y$ measurements, Alice is able to produce and maintain volume-law entangled states at late times. We now investigate if this ability also allows her to protect a quantum diary. As in section III B, we investigate Alice's ability to protect a quantum diary of $|A| = L$ qubits using the purification dynamics of an initial mixed state

with with entropy $S(\rho_0) = L$. At late times, the entropy of the system is again equivalent to the coherent information between the initial and final system states, $S(L) = \mathcal{C}$. Thus, when the system purifies, Alice looses her diary, while when it remains mixed, Alice can recover $S(L) = \mathcal{C}$ qubits of her diary. We consider a circuit with $p_m = 0.03$ and $p_y = 1/4$ fixed, and study its purification dynamics as a function of $\Delta_x$. In Fig. 10, we observe a transition between the protection and loss of the quantum diary occurring at a critical point $\Delta_x \approx 1/3$, the point at which Alice gains access to volume coherence in the $p_m \to 1/L^2$ limit. Thus, the ability of Alice to protect a quantum diary, is accurately predicted from the condition that Eve's measurements give Alice access to large coherence, $C_x > L/2$. This provides evidence for a connection between quantum communication and coherence. In Appendix E, we give a first numerical analysis of the critical behaviour of the transition presented in this section, but leave for future work a detailed investigation into the critical dynamics induced by the complex interplay between coherence and entanglement. Finally, we note that the usual measurement induced transition, tuned $p_m$, remains so long as $p_y > |p_x - p_y|$.

## V. COHERENCE REQUIREMENTS FOR QUANTUM COMMUNICATION

### A. Alice's coherence generating requirements

In sections III and IV we found that if Alice is constrained in her ability to generate coherence, then she can only protect quantum information if Eve is restricted in her ability to destroy coherence. In section III, we found that Eve must not apply coherence-destroying bit erasers, while in section IV she must apply a sufficient ratio of $Y$ measurements to $X$ and $Z$ measurements $p_y > |p_x - p_z|$. Thus, if Eve is able control which operator she makes a measurement of, she could choose to always measure the $X$ basis ($p_x = 1$) and quickly destroy $X$ coherence and Alice's quantum diary. Alice is therefore required to dynamically regenerate coherence in the system as it evolves. A simple way of doing this is by adding a finite rate of phase gates $R_z$ which can generate coherence in the $X$ basis. We now show that this solution works when random strategies are played and allows Alice to protect against the coherence-maintaining bit flip errors and coherence destroying $X$ measurements.

The dynamics for a fixed error rate $p_m + p_e = 0.05$, with $p_y = p_z = 0$ and $p_x = 1$, are displayed in Fig. 11, and show that for a sufficient rate of phase gates $p_R > 0.1$ the system can protect any ratio $p_m/p_e = r_d/(1 - r_d)$ of $X$-measurements (dephasing errors) to bit erasers. For $p_R < 0.1$, the system can protect only a fixed fraction of bit flip errors and this fraction is related to when the steady state coherence reaches $C_x \approx L/2$ (also shown in Fig. 11). This correspondence demonstrates the necessity of quantum coherence to maintain a finite quantum channel capacity.

a)

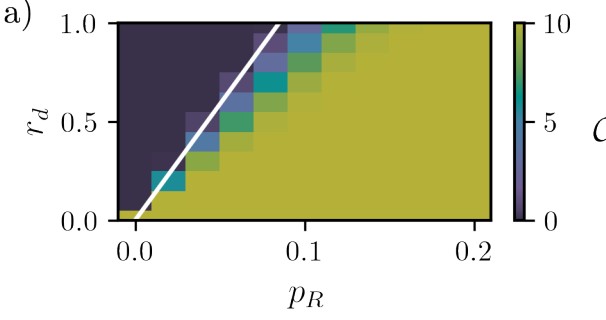

b)

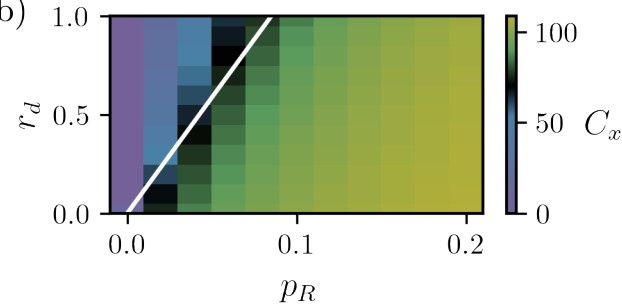

Figure 11. Quantum channel capacity $\mathcal{C}$ for a diary encoding $|A| = 10$ qubits (panel a) and $X$ basis coherence $C_x$ (panel b) at time $t = 10$ for a circuit composed of CNOTs, single bit phase gates occurring at a rate $p_R$, and two types of attack operations occurring at a rate $p_m + p_e = 0.05$: $X$ measurements occur at a rate $p_m = 0.05 * r_d$ (i.e. $p_x = 1$ and $p_y = p_z = 0$) while coherence-maintaining bit erasers occur at a rate $p_e = 0.05(1-r_d)$. Panel a) demonstrates a transition between zero and finite quantum channel capacity for sufficiently large rate of coherence generating phase gates. Panel b) shows that this transition is associated with a sharp change in coherence $C_x$. The color scheme in the panel b) is chosen such that $C_x = L/2 = 64$ is black.

Furthermore, this demonstrates that coherence can be preserved simply by the addition of a sufficient rate of single qubit coherence generating gates.

### 1. Coherence requirements for any of Alice's strategies

We now consider the setting in which Alice does not take a random strategy, but instead can make any specific choice of unitaries from the Clifford group. We will also assume that Alice takes turns with Eve in applying their choice of operations, and that Eve is allowed to apply at most an integer number, $M$, of local Pauli measurements in her turn. Again, we assume Alice knows the basis Eve makes measurements and also the outcome of those measurements.

The coherence requirements on Alice are different depending on whether Eve declares where she will make measurements at the beginning of her turn or not. When Eve does declare this, Alice can encode a diary of $L - M$ bits with only swap operations (incoherent operations in

any local basis). She achieves this by encoding her state on $L - M$ of the bits and during her turn swapping the remaining $M$ bits to the locations Eve will measure. If Eve does not announce where she will measure, Alice will have to ensure a measurement on any qubit won't measure her encoded state and reduce her channel capacity. Equivalently in the purification picture [26], Alice must ensure that Eve can not reduce the evolved purity below her desired channel capacity $S(\rho_n) \geq \mathcal{C}$. For $\rho_n$ taken as a stabilizer state, the condition for a Pauli measurement to reduce the entropy of a state $\rho_n$ translates to a condition on the coherence of the evolving density matrix $\rho_n$. In Appendix B, we derive this condition as the following theorem:

**Theorem 2.** *Given a local Pauli basis D, any stabilizer mixed state $\rho$ with entropy $S(\rho)$ and relative entropy of coherence $C_D(\rho)$, there exists a sequence of $M > C_D(\rho)$ local Pauli measurements that reduce the entropy of $\rho$ (the state after measurement, $\rho'$, has entropy $S(\rho') < S(\rho)$).*

Thus, if Alice is unable to maintain a coherence of $C_x(\rho_n) \geq M_x$ in her circuit, then Eve can use the $M_x$ measurements guaranteed by Theorem. 2, to reduce the entropy $S$ of Alice's mixed state. If Alice is limited in this way, then Eve can, in at most $L$ steps purify Alice's state and reduce the quantum channel capacity to zero. Since Eve could make a measurements in any local Pauli basis, then Alice must maintain a coherence $C_D > M$ in all local Pauli bases $D$. This holds for any strategy Alice might take, not knowing where Eve will apply her measurements.

### B. Coherence requirements for stabilizer error correction codes

In the previous sections we observed that on the one hand, Alice can protect classical information by maintaining a classical code space with a large Hamming distance between the code words of the classical code space. On the other hand, we observed that for Alice to protect quantum information, she was required to maintain a large amount of superposition between different basis states (i.e. a large volume-law coherence). In classical error correction codes, the Hamming distance between any two code words is related the number of bit flip errors that can be corrected (i.e. the code distance) [72]. This, along with previous results studying the role of quantum coherence in channel discrimination tasks [18–23], suggests quantum coherence is related to the code distance of quantum codes. In this section, we formalize these expectations by presenting Theorem 3 that states the code distance of any stabilizer quantum error correction code is bounded by the coherence of its maximally coherent state for any local Pauli basis. We then explain how this bound can be made tighter by considering different sub-codes, and discuss the relevance of this bound to the information game discussed in the previous section. Fi-

nally, we conclude this section with an application of the bound to CSS codes.

The theorem applies to $[[N, k, d]]$ stabilizer codes that use $N$ qubits to detect up to $d$ errors on a quantum code space of dimension $2^k$. The code space, $P$, is defined by a set of $N - k$ Pauli check operators $\{g_i\}$ that constrain the states that can live in the code space, $|\psi\rangle \in P$, by the constraint $g_i |\psi\rangle = |\psi\rangle$ for all $i \in (1 \ldots N - k)$.

**Theorem 3.** *Given a local Pauli basis $D$, the code distance $d$ of a $[[N, k, d]]$ stabilizer code, $P$, is bounded by the coherence of the maximally coherent stabilizer state in the code space:*

$$d \leq \max_{\psi \in P} C(|\psi\rangle, D) \equiv C_{PD}. \quad (13)$$

Here, a local Pauli basis $D$ is any basis that, on each site $i$, one of the Pauli operators $A_i \in (X_i, Y_i, Z_i)$ is diagonal. A proof of this theorem is given in Appendix C and is constructed by identifying an undetectable error composed of $d = C_{PD}$ measurements of a subset of the Pauli operators $A_i$ defining the Pauli basis $D$. Intuitively, such an error can be constructed because any state in the code space has at most $C_{PD}$ coherence. Therefore, the coherence of such a state can be reduced to 0 in at most $C_{PD}$ dephasing errors (measurements), thus destroying all phase information the state might have encoded in the basis states of the Pauli basis $D$.

Notice that, while this bound is expressed in terms of the maximum coherent stabilizer state of the code space, it is actually tighter. The tighter bound can be obtained by applying the theorem to any subspace of the code space. Then, since an error for any subspace of the code space is an error for the whole code space, the coherence of the maximum coherent state of the subspace bounds the code distance. This way the bound is actually closer to the coherence of the second least coherent stabilizer code state. This is seen by constructing a basis of stabilizer states which span the full code space and ordering them by the coherence. The tightest bound comes from the subspace formed from the two least coherent stabilizer basis states. Within this subspace, the maximum coherence is at most 1 bit more than the second least coherent basis state, and thus the tightest bound is obtained using that basis state.

### 1. Application to Alice-Eve information game

Applying such a bound to the game played between Alice and Eve is difficult because there is no static code space that we can apply the bound to. Nonetheless, we can use the intuition from $C_{PD} \geq d$ to understand why the transition line in Fig. 11 is linear. Our approach will first identify an effective code distance, $d_{eff}$; then it will provide an estimate on the relevant coherence, $C_{PD}$, limiting Alice's ability to protect her diary; and finally use $C_{PD} = d_{eff}$ to identify the critical line. Thus, we first note that the error rate, $p_m + p_e$, generated by Eve

is fixed in the $p_m$ versus $p_R$ plane of Fig. 11. This allows us to assume the maximum number of errors that might need to be corrected is some constant $d_{eff} = K$ that does not depend on $p_m$ or $p_R$. Then, as argued in the previous section, the coherence in the $X$ basis was limiting Alice's quantum channel capacity and so we derive the $p_m$ v.s. $p_R$ phase boundary using $C_{PD} = C_x(p_m, p_R) = K$: when the steady state coherence in the $X$ basis $C_x(p_m, p_R)$ is sufficiently large, $C_x(p_m, p_R) \geq d_{eff} = K$, then Alice can protect her quantum diary, otherwise, she can't and her channel capacity drops to zero. Following similar arguments as in Appendix D 3, we propose the following rate equation for $C_x$:

$$\partial_m C_x = p_R - p_x \left( 1 - \left( \frac{N_x}{L} \right)^\xi \right). \quad (14)$$

Solving for the steady state, we find that the critical phase gate rate is given by $p_R = p_x \left[ 1 - (1 - K/L)^\xi \right]$, and for a fixed $L$ we find the critical phase gate rate scales linearly with $p_x$ as observed in Fig. 11. For increasing $L$, the number of errors per time step increases linearly with $L$ such that the effective code distance, $K$, should also scale linearly with $L$. Thus, the proportionality of the critical line $p_R \sim p_x$ does not change with increase $L$ as we have confirmed numerically.

### 2. Application to the L-bit repetition code

The application to such a bound for a stabilizer error correction code is much simpler than for the information game since the theorem now applies directly. We first consider the simplest quantum error correction code, the repetition code [72, 79] which is the quantum generalization of the simple classical coding procedure of repeating a message multiple times. The code space for the $L$-bit repetition code is spanned by the $X$-basis states $|00 \cdots 0\rangle$ and $|11 \cdots 1\rangle$, and can correct up to $(L - 1)/2$ bit flip errors but zero $X$-dephasing errors. The application of the theorem is done by considering the state $(|00 \cdots 0\rangle + |11 \cdots 1\rangle)/\sqrt{2}$ which has maximum $X$ coherence of $C_x = 1$ in the code space. Thus $d \leq 1$ and the repetition code can correct up to $(d - 1)/2 = 0$ $X$-dephasing errors. While the fact the repetition code can not correct phase errors is obvious, the application of Theorem 3 shows that this inability is because the code lacks the ability to produce states with coherence in the $X$ basis.

### 3. Application to CSS codes

While the above examples are rather simple, they show coherence provides a unifying and general view for certain requirements when designing quantum error correction codes. In this section, we show that this general context provided by the coherence bound is related to

the generality of the singleton bound for classical error correction codes. The classical Singleton bound [72] is a bound $d \leq L - k + 1$ on the code distance, $d$, for any classical code determined strictly from the size of the code space $2^k$ and the number of physical bits used in the code, $L$. The relation to the coherence bound comes from applying Theorem 3 to the CSS quantum error correction codes which are codes constructed using two classical codes $\mathcal{K}^x$ with an $k_x$ bit code space, and $\mathcal{K}^z$ with a $k_z$ bit code space defined on $L$ physical bits. These codes compose a large class of stabilizer error correction codes [72, 99], and are useful for constructing codes will good asymptotic properties [100].

To apply the coherence bound, we recall that a basis for the CSS quantum code space can be constructed by using the dual code $\mathcal{K}_\perp^z$ with code space of size $L - k_z$. The code words, $x$, of the dual code are the bit strings of length $L$ generated by the transpose of parity check matrix $H_{ij}^z$ of the code $\mathcal{K}^z$:

$$x = G_\perp^z z = (H^z)^T z \tag{15}$$

for all bit strings $z$ of length $k^z$. Using these code words of the dual code, $\mathcal{K}_\perp^z$, the $2^k = 2^{k_x - (L - k_z)}$ distinct stabilizer basis states for the CSS code can be easily written [72] in the $X$ basis as:

$$|x + \mathcal{K}_\perp^z\rangle = \frac{1}{\sqrt{|\mathcal{K}_\perp^z|}} \sum_{y \in \mathcal{K}_\perp^z} |x + y\rangle. \tag{16}$$

for all distinct $x \in \mathcal{K}^x / \mathcal{K}_\perp^z$. This shows immediately that the coherence of each basis state is $C_x = \log_2(|\mathcal{K}_\perp^z|) = L - k_z$. We can then bound the code distance by considering the maximal coherent state for a single bit logical subspace spanned by states $|x + \mathcal{K}_\perp^z\rangle$ and $|x' + \mathcal{K}_\perp^z\rangle$ such that $x + y \neq x'$ for all $y$ in $\mathcal{K}_\perp^z$. The maximum coherent state of this subspace is the superposition $|\psi\rangle = (|x + \mathcal{K}_\perp^z\rangle + |x' + \mathcal{K}_\perp^z\rangle)/\sqrt{2}$ which has coherence $C_x = L - k_z + 1$. Thus the coherence bound for the code distance is $L - k_z + 1 > d$ which retrieves the classical Singleton bound of $\mathcal{K}^z$ which is used to correct phase errors [72]. A similar analysis in the $Z$ basis will retrieve the classical Singleton bound for the $\mathcal{K}^x$ code space with $L - k_x + 1 > d$.

## VI. DISCUSSION AND OUTLOOK

In this work, we have determined the coherence resource requirements for quantum communication, both for stabilizer error correction codes and for monitored quantum dynamics modeled by an information game. The game involved Eve applying measurements in an attempt to destroy Alice's quantum diary, and Alice recording the measurements (their location, polarization and outcome) and applying unitaries in an attempt to protect her diary. By considering the purification dynamics of the associated circuits, we determined Alice's ability to protect her diary and found that her access to coherence

or coherence generating operations allowed her to protect either classical information or quantum information. When Alice is limited in her access to coherence, she can only protect classical information, while if she has access to coherence or coherence generating operations she can protect quantum information.

This is summarized in Fig. 12, which shows a figure similar to Fig. 11 in which a purification transition in Alice's quantum channel capacity depicted. In contrast to Fig. 11, we show what type of information she is capable of protecting given a certain type of attack by Eve and a maximum rate, $p_R$, at which she can apply a coherence generating operations. The main difference is that, in Fig. 12, we treat $p_R$ as a limit on her ability to generate coherence, rather then the rate she actually applies the coherence generating operation (for instance, a phase gate). This is relevant because when Alice applies a phase gates, she generate quantum noise in the $X$ basis states, which looks like an error from the perspective of the classical channel. Specifically, the classical channel capacity is the mutual information between an initial distribution of $X$ basis states and the final distribution of $X$ basis states. Thus, a phase gate is a quantum error because it generates quantum uncertainty (i.e. coherence) in the $X$ basis states. As a trivial example, if Alice first prepares a $X$ basis state on a qubit, then applies first a phase gate, and finally makes a measurement in the $X$ basis, the final state of the qubit will be uncorrelated with the initial state. The effect on classical channel capacity due to the application of phase gates was more deeply investigated in Ref. [76].

Thus, the circuits in Fig. 11 have zero classical channel capacity in the whole phase diagram, because Alice is applying phase gates. If instead, she were to not apply the phase gate and attempt to encode classical information, she could protect a classical diary. Therefore, in Fig. 12, we imagine $p_R$ as the rate at which she could apply phase gates, and we observe the transition is from an ability to protect classical information to an ability to protect quantum information.

### A. Generality

While here, we have considered circuits composed only of Clifford circuits, we expect our results to hold when the coherence-generating operation is chosen from a universal gate set. While above, we only considered a phase gate as the coherence-generating gate, many other gates can play a similar role. In fact, a two-qubit gate, chosen randomly using the Haar measure, produced, on average, $\sum_{k=2}^{4} 1/k \approx 1.08$ bits of relative entropy of coherence in any basis when acting on a pure state [101]. Similarly, phase gates produce 1 bit of relative entropy of coherence in the $X$ basis when acting on an $X$-polarized qubit, and thus, it seems natural that a similar transition to that discussed in section V A would occur if phase gates are replaced by randomly chosen two-qubit gates. The main

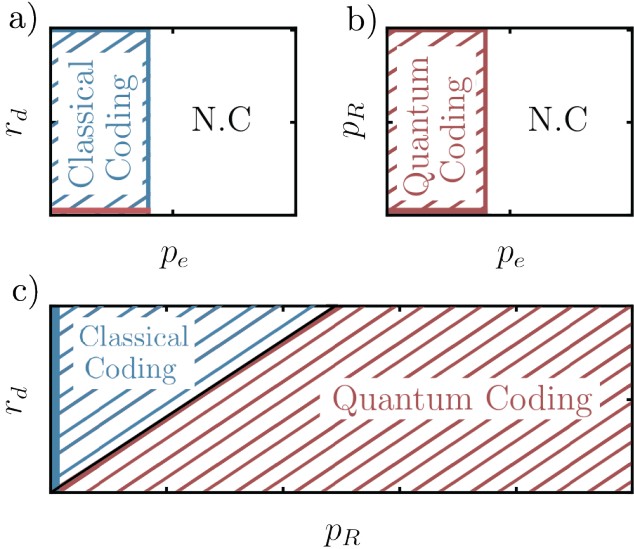

Figure 12. Alice's coding capabilities in various regions of parameter space with $p_z = p_y = 0$ and $p_x = 1$. In all figures, we consider Eve's attacks to be coherence-maintaining bit erasers (at a rate $p_e$) and $X$ measurements (at a rate $p_m = p_e r_d/(1 - r_d)$), and we give Alice the option to apply phase gates up to a rate $p_R$. a) For $p_R = 0$, Alice can only encode a quantum diary if Eve does not apply coherence destroying measurements (red line at $r_d = 0$), otherwise Alice can protect a classical diary as long as $p_e$ is sufficiently small. b) In this figure $p_m = 0$ and so Alice can always protect a quantum diary as long as she can protect a classical diary. When $p_R = 0$, we retrieve the limit discussed in section III B where only coherence-maintaining bit erasers are applied. c) In this figure $p_e = 0.05 - p_m$ as in Fig. 11, but now Alice has the choice to apply phase gates or not. The figure shows the transition to quantum error correction as in Fig. 11, but now if she chooses not to apply phase gates she can protect classical information so long as $p_e < 0.1$.

difference will be that the fluctuations of coherence will be different, and the critical properties of the phase transition will change, similar to previously studied random circuits [32, 34, 36, 98]. Testing this expectation and investigating the difference in fluctuations could be a fruitful direction for future work.

Future research might also find it interesting to study the coherence requirements of Alice in the setting in which she can apply non-Clifford gates. We suspect a similar bound as in Theorem 2 to hold. Intuitively, this theorem is related to the fact that a quantum uncertainty of $C_D(\rho)$ bits, of the basis states $D$, can be reduced to zero by $C_D(\rho)$ measurements on the basis $D$. For stabilizer states, this uncertainty is distributed evenly across $2^{C_D(\rho)}$ basis states, so the theorem follows naturally. For non-stabilizer states, the uncertainty is distributed unevenly across potentially more states. In this case, a rare measurement outcome may increase or decrease the number of measurements required to reduce the quantum uncertainty. It is therefore likely that the bound only holds

on average or in the limit of a repeated number of states, $n$, whose quantum uncertainty is reduced after $nC_D(\rho)$ measurements. The alternative is that magic [102, 103], a resource quantifying how much a state is not a stabilizer state, can substitute the resource of coherence in this setting. It would, therefore, be interesting for future work to confirm this intuition. Here, it is important to note that all of our results hold for coherence in a local basis, and we do not expect our results to easily generalize to coherence in a global basis.

It could also be interesting to study the role of coherence in bases specified by a symmetry relevant to the unitary dynamics. Specifically, one might consider a generalization of the dynamics investigated in Refs. [56, 57, 69, 104], which studied dynamics that preserved a global symmetry. One could instead include unitaries that break the symmetry but that are still incoherent operations in a basis labeled by the conserved charges. In this case, coherence between different charge sectors would remain suppressed, but the dynamics would allow for classical fluctuations of the conserved charge.

### 1. Possible speed up for classical simulations

Another interesting setting, is to consider circuits that simulate a given Lindblad unraveling of a specific open system dynamics. In this case, there could also be an entanglement transition tuned by the relative rate at which the unitary part of the dynamics generates coherence and the dissipative part destroys coherence. In this case the channel capacity of the specific unraveling won't be particularly interesting, but the entanglement transition from area law to volume law entangled states would be a transition in the ability of classical computers to simulate the open system dynamics [47, 50, 81]. In this setting, our results suggest a heuristic for simulating Lindblad dynamics: use an unraveling which minimize the coherence in a given local Pauli basis. By doing this, entanglement growth will be suppressed possibly allowing for longer classical simulations using matrix product states (MPS). As shown in Section IV, certain dynamics can show a transition to area law entanglement for arbitrarily weak coupling to an environment (measurement rate). If such a transition occurred by choosing between two different unravelings of the Lindblad, then this heuristic would lead to an exponential speed up by choosing the right unraveling: In one unraveling, entanglement would growth linearly in time, and classical representations of the late time states would require memory growing exponentially in system size. While in another unraveling, suppression of coherence would ensure late time area law entanglement, such that memory requirements would only grow linearly in system size.

## B. Perspective on the entangling phase

From a different perspective, this transition in the capacity to protect either quantum or classical information provides an answer to what is quantum about the entangling phase of the measurement-induced entanglement transitions. This question is raised by transitions in classical information quantifiers observed in the classical circuits of section III, and Refs. [75–78]. One answer was given in Ref. [76], which showed quantum gates (i.e. quantum noise), such as the phase gate, are an instability to the ordered classical phases. Another answer is given by Fig. 11, which shows that in the quantum phase, quantum information is protected from both bit and phase errors (coherence destroying errors).

Thus, the answer to what is quantum about the measurement-induced entanglement transitions is equivalent to the question, what is quantum about quantum error correction codes. In both cases, it is a stability to both bit and phase errors. Also, in both cases, the extent to which both are stable to $X$ or $Z$ errors corresponds to the amount of $X$ or $Z$ coherence respectively. It is only when the phase, or quantum code space, contains states with both $X$ or $Z$ coherence will they be stable to $X$ or $Z$ errors. In the case of measurement-induced phase transitions, this fact is reflected in a volume-law to area-law phase transition corresponding to a critical loss of either $X$ or $Z$ coherence (cf. Section IV B). While in the case of quantum error correction codes, the Singleton bound on the code distance for $X$ or $Z$ bit errors in a CSS code corresponds to the coherence bound on the code distance applied to either the $X$ or $Z$ basis.

## C. Future directions

The coherence bound may also provide a useful design principle for quantum error codes targeting a given experiment. For example, while generating coherence in any given basis can be achieved by single qubit operations, generating it for all local Pauli basis requires entangling gates and is therefore more difficult. Thus experiments will generally be limited in the amount of coherence they generate in some basis. By identifying this limit on coherence, one could identify a maximum code distance the experiment could produce, and focus on constructing codes with a code distance less than that.

Our work therefore provides interesting directions in both the design of systems protecting quantum information and the relation between classical and quantum information dynamics. While here, we have studied the classical limit of quantum gate operations, it could also be interesting to study how coherence brings one away from the classical limit of chaotic dynamics of continuous systems. In the present work, spreading and scrambling of quantum information was done by controlled gate operations, but one may also be interested in the natural scrambling of information present in both classical and quantum chaotic systems [105–117]. Since here coherence distinguished between classical and quantum dynamics, it might also be possible that it can elucidate results connecting classical to quantum chaos [118–120]. Finally, one may also think to use random circuits to investigate how resources quantified by other resource theories [4], like asymmetry, non-gaussianity, contextuality or incompatability are required for quantum or classical communication. A similar procedure taken here might be useful but where Alice and Eve are instead constrained to use the free operations of a different resource theory. In conclusion, our results can provide a new perspective on the connection between classical and quantum information scrambling, and their relation to communication technologies.

*Acknowledgments*– This work has been funded by the Deutsche Forschungsgemeinschaft (DFG, German Research Foundation) - TRR 288 - 422213477 (project B09), TRR 306 QuCoLiMa ("Quantum Cooperativity of Light and Matter"), Project-ID 429529648 (project D04) and in part by the National Science Foundation under Grant No. NSF PHY-1748958 (KITP program 'Non-Equilibrium Universality: From Classical to Quantum and Back'), and by the Dynamics and Topology Centre funded by the State of Rhineland Palatinate and Topology Centre funded by the State of Rhineland Palatinate. M.P.A.F was supported by the Heising-Simons Foundation and by the Simons Collaboration on Ultra-Quantum Matter, which is a grant from the Simons Foundation (651457). The authors gratefully acknowledge the computing time granted on the supercomputer MOGON 2 at Johannes Gutenberg-University Mainz (hpc.uni-mainz.de).

## VII. APPENDIX

### Appendix A: Properties of stabilizer states

**Definition 1.** *A **stabilizer mixed state**, $\rho$ on $L$ qubits, with von Neumann entropy $S(\rho)$ is defined using a group $S$ generated by a set of $N_s = L - S(\rho)$ operators which are strings of single site Pauli operators $\{g_i\}$:*

$$\rho = \prod_{i=1}^{L-S(\rho)} \frac{1+g_i}{2} \tag{A1}$$

We use the stabilizer check matrix, $C_{ij}$ which is a $L-S(\rho)$ by $2L+1$ matrix, where the rows index the generators and the columns specify the form of the generator:

$$g_i = (-1)^{C_{i,2L+1}} \prod_{j=1}^{L} X_j^{C_{i,j}} \prod_{j=L+1}^{2L} Z_j^{C_{i,j}}. \tag{A2}$$

The stabilizer check matrix has entries from the field $\mathcal{F}_2 = (0,1)$, and act in a vector space defined over that

field, where multiplication and addition are performed modulo 2. The set of all Pauli string operators that commute with the elements of $S$ are called the *centralizer* of $S$. This set forms a group $C(S)$ where $p \in C(S)$ if $pg_i = g_i p$ for all $g_i \in S$. It will be important to consider the set, $C - S$, of Pauli strings operators contained in the centralizer but not contained in the stabilizer group.

### 1. Representations of stabilizer states

A stabilizer mixed state is defined by its stabilizer group, $S$, and can have different representations based on the different choices of generators $g_i$ of that group [72]. Changes between representations, often called gauges, can be made by changing a generator, $g_i$ by group multiplication

$$g_i \rightarrow g_i' = Rg_i, \tag{A3}$$

where the group element $R \in S/g_i$ is taken from the group, $S/g_i$, generated by all the generators of $S$ not including $g_i$. If we write $R = (-1)^{r_{2L+1}} \prod_{j=1}^{L} X_j^{r_j} \prod_{j=L+1}^{2L} Z_j^{r_j}$, then such a procedure changes the stabilizer check matrix, by adding the vector $r$ to the $i^{th}$ row of the stabilizer check matrix $C_{ij}$ (again all operations performed modulo 2) [72]. Thus different representations of the stabilizer state correspond to different stabilizer check matrices all related to each other by row operations. Gaussian elimination will be a useful tool to find convenient representations of the stabilizer state when proving lemma 5.

### 2. Measurements on stabilizer states

The measurement of a Pauli string, $O$ on a stabilizer mixed state, can have three possible effects:

- No effect: Occurs when $\pm O \in S$, since the stabilizer state is already an eigenstate of the measurement operator $O$.

- $O$ changes the state and reduces the entropy. In this case, $O$ is added to the generators of the stabilizer state, and occurs when $O$ in the centralizer of $S$ but not in $S$: $O \in C(S)$ and $O \notin S$.

- $O$ changes the state, but not the entropy. This case occurs when $[O, g_i] \neq 0$ for at least one of the generators $g_i$ and it requires a non trivial update to the stabilizers state.

In the last case, updating the state is performed [72] by:

1. Changing the representation of the stabilizer state to one in which only one generator $g_1'$ does not commute with $O$.

2. Replacing the generator $g_1'$ with the new generator $O$.

### 3. Coherence of stabilizer states

We now prove the lemmas and theorem discussed in the text, and in particular, we prove Theorem 6 which gives an expression for the relative entropy of coherence for a stabilizer state. To be as general as possible, it is useful for us to define

**Definition 2.** *A **local Pauli basis** is as any basis that, on each site $i$, one of the Pauli operators $A_i \in (X_i, Y_i, Z_i)$ is diagonal.*

The basis states, $\{|s\rangle\}$, of a local Pauli basis are defined by a given bit string $s = \{\alpha_1, \alpha_2 \cdots \alpha_L\}$ obtained after performing a projective measurement on $L$ Pauli operators $\{A_i\}$:

$$A_i |s\rangle = (-1)^{\alpha_i} |s\rangle \tag{A4}$$

where $\alpha_i \in (0, 1)$. We will also define $H(p(x)) = -\sum_x p(x) \log(p(x))$ to be the Shannon entropy of a distribution $p(x)$.

We will often find it useful to use a generalization of the following property of measuring a stabilizer states discussed in appendix A 2: the outcome of measuring a Pauli operator, $A_i \in (X_i, Y_i, Z_i)$ is either certain ($P(\alpha_i = 0) = 1$ or $P(\alpha_i = 1) = 1$), or uncertain with probability $P(\alpha_i) = 1/2$, where $\alpha_i \in (0, 1)$ defines an eigenvalue, $(-)^{\alpha_i}$, of $A_i$. When applied to a sequence of measurements on the Pauli operators $\{A_i\}$ defining a Pauli basis, we obtain the following theorem:

**Lemma 4.** *Given a stabilizer mixed state $\rho$, the probability, $P(s)$ for the bit string, $s$, resulting from a measurement on a local Pauli basis $A$, is uniform over $2^{n_u}$ bit strings where $n_u = H(P(s))$ is the number of uncertain measurement outcomes in a sequence of measurements $A_1, A_2 \ldots A_L$. i.e. $P(s) = 2^{-n_u}$ if the bit string is one of the $2^{n_u}$ allowed bit strings or $P(s) = 0$ if not.*

*Proof:* From Born rule, the probability:

$$P(s) = \text{Tr}\left[\prod_i D(A_i = \alpha_i)\rho\right] = \text{Tr}\left[\frac{(\prod_i(-1)^{\alpha_i} A_i + 1)}{2}\rho\right]$$

where $A_i$ is the single site Pauli operator on the $i^{th}$ site defining the local Pauli basis $A$, and $\prod_i D(A_i = \alpha_i)$ is the projector to the basis state $|s\rangle$. We can compute this probability $P(s)$ sequentially by performing a sequence of projective measurements on the $i^{th}$ site:

$$\rho_i = D(A_i = \alpha_i)\rho_{i-1}D(A_i = \alpha_i) \tag{A5}$$

where $\rho_0 = \rho$ and $\rho_L = P(s)|s\rangle\langle s|$ is the projected Pauli basis state with normalization $P(s) = \text{Tr}(\rho_L)$.

The probability distribution $P(s)$ will depend on whether the probability distribution of the individual measurements

$$p_i(\alpha_i) = \text{Tr}[D(A_i = \alpha_i)\rho_{i-1}D(A_i = \alpha_i)]/\text{Tr}[\rho_{i-1}]. \tag{A6}$$

are certain or uncertain. There are four options for the outcome of the $i^{th}$ projection on the $i^{th}$ stabilizer state with stabilizer group $S_i$:

1. $A_i \in S_i$; $p_i(\alpha_i) \in (1, 0)$ and the measurement outcome is certain. In this case $\rho_i = \rho_{i-1}$ or $\rho_i = 0$ depending on $\alpha_i = 0$ or 1 respectively, and the measurement outcome of $A_i$ is certain.

2. $-A_i \in S_i$; $p_i(\alpha_i) \in (1, 0)$ and the measurement outcome is certain. This is the same case as 1) but projection onto a different outcome $\rho_i = \rho_{i-1}$ if $\alpha_i = 1$ and $\rho_i = 0$ if $\alpha_i = 0$.

3. Both $A_i \in C(S_i) - S_i$ and $-A_i \in C(S_i) - S_i$; $p_i(\alpha_i) = 1/2$ and the measurement outcome is uncertain: in this case $\rho_i = D(A_i = \alpha_i)\rho_{i-1}D(A_i = \alpha_i) = D^2\rho_{i-1} = D\rho_{i-1}$ independent of $\alpha_i$. Here $D(A_i = \alpha_i) \equiv D$.

4. $A_i \notin C(S_i)$; $p_i(\alpha_i) = 1/2$ and the measurement outcome is uncertain. In this case, we can choose a representation of $S_i$ such that only one generator $g_j$ does not commute, $[g_j, A_i] \neq 0$. Furthermore, since $g_j$ and $A_i$ are non commuting Pauli operators we have $g_j A_i = -A_i g_j$, and can compute:

$$\frac{((-1)^{\alpha_i} A_i + 1)(1 + g_j)((-1)^{\alpha_i} A_i + 1)}{8} \quad \text{(A7)}$$
$$= \frac{1 + (-1)^{\alpha_i} A_i}{4}$$

such that the state $\rho_i$ is the stabilizer state with the $j^{th}$ term in the product of Eq. A1 replaced by $(1 + (-1)^{\alpha_i} A_i)/2$ and normalization such that $p_i(\alpha_i) = 1/2$.

We can then find the probability of a measurement outcome $P(s)$ from the probability of the $i^{th}$ measurement outcome $p_i(\alpha_i)$:

$$P(s) = \text{Tr}[\rho_L] = p_L(\alpha_L) \text{Tr}[\rho_{L-1}] = \prod_{i=1}^{L} p_i(\alpha_i) \quad \text{(A8)}$$

which is either $2^{-n_u}$ or 0 where $n_u$ is the number of uncertain measurement outcomes in the sequence. Furthermore, whether a measurement outcome is certain (case 1 and 2) or uncertain (case 2 or 3) depends only on the measurement being performed, $A_i$, and not the previous measurement outcome, $\alpha_j$ for $j \leq i$. Thus we find that regardless of the bit string being projected $s$, the number of uncertain outcomes is the same. We conclude that $P(s)$ is a uniform distribution (for all $P(s) \neq 0$ we have $P(s) = 2^{-n_u}$) with entropy $H(P(s)) = n_u$ $\square$.

Notice that since this gives us the entropy of $P(s)$ such that we can easily compute the relative entropy of coherence in the basis $A$ as:

$$C(\rho, A) = H(P(s)) - S(\rho) = n_u + N_s - L \quad \text{(A9)}$$

where $N_s$ is the number of independent generators in the stabilizer group $S$.

To easily obtain the number $n_u$ of uncertain outcomes, we find the following stabilizer representation useful:

**Lemma 5.** *The CSS "gauge"* *For a given stabilizer mixed state, there exists a representation, called the CSS gauge, in which the stabilizer check matrix takes the following form:*

$$\begin{bmatrix} G_x^x & 0 & s^x \\ 0 & G_z^z & s^z \\ G_y^x & G_y^z & s^y \end{bmatrix} \quad \text{(A10)}$$

*where the rows of $G_y^x$ are linearly independent, and where the rows of $G_y^z$ are also linearly independent. In Eq. B1, $G_x^x$ is a $N_x$ by $L$ matrix defining, along with the column $s^x$, a set of generators $\{g_i^x\}$ composed solely of $X_i$ operators; $G_z^z$ is a $N_z$ by $L$ matrix defining, along with the column $s^z$, a set of generators $\{g_i^z\}$ composed solely of $Z_i$ operators; while $G_y^x$ and $G_y^z$ are $L - S(\rho) - N_x - N_z$ by $L$ matrices that together with the column $s^y$ define a set of generators $\{g_i^y\}$.*

In this gauge, there are a set of $N_x$ generators $\{g_i^x\}$ defined via the matrix $G_x^x$ composing only $X$ Pauli strings: $g_i^x = (-1)^{s_i^x} \prod_j X_j^{(G_x^x)_{i,j}}$. Similarly there are a set of $N_y$ generators $\{g_i^z\}$ defined via the matrix $G_z^z$ composed only of $Z$ Pauli strings, while the remaining $L - S(\rho) - N_x - N_z$ generators are products of both $X$ and $Z$ Pauli operators.

*Proof:* We first note that, as discussed in appendix A 1, row operations applied to the stabilizer check matrix correspond to multiplication of the generators $g_i$ by elements of $S/g_i$, such that different representations of the stabilizer group $S$ are related by row operations applied to the stabilizer check matrix. The proof then proceeds by construction. First start with a generic stabilizer check matrix

$$\begin{bmatrix} g_1^x & g_1^z & s^1 \end{bmatrix} \quad \text{(A11)}$$

Where $g_1^x$ and $g_1^z$ are $L - S$ by $L$ matrices. If $g_1^z$ has $M_z < L - S(\rho)$ linearly independent rows, then by Gaussian elimination on the columns $j = L + 1 \dots 2L$, will obtain a new matrix with

$$\begin{bmatrix} g_x^x & 0 & s^x \\ g_2^x & g_2^z & s^2 \end{bmatrix} \quad \text{(A12)}$$

where $g_2^z$ has $M_z$ linearly independent rows and $g_x^x$ has $N_x = L - S(\rho) - M_z$ rows. Furthermore since the generators are independent and do not contain $I$ or $-I$, the rows of $g_x^x$ must be linearly independent such that the check matrix can not contain a row with all 0s in the columns $1 \dots 2L$. Now the combined set of rows from both $g_x^x$ and $g_2^x$ may also have only $M_x < L - S(\rho)$ linearly independent rows, such that Gaussian elimination can eliminate $N_z = L - M_x$ rows of $g_2^x$. Applying that Gaussian elimination on Eq. A12 we obtain the CSS gauge Eq. B1 $\square$.

We can now derive the relative entropy of coherence of Stabilizer states for the coherence in the $X$ and $Z$ basis's:

**Theorem 6.** *The coherences in the $X$ and $Z$ basis of a stabilizer state are determined by number the of rows $N_x$, $N_z$ and $N_y$ of the matrices $g_x^x$, $g_z^z$ and $g_y^x$ of the CSS gauge:*

$$C_x = N_y + N_z \qquad \text{(A13)}$$
$$C_z = N_y + N_x$$

*Proof:* The coherences are given as $C_x = H(P(x)) - S(\rho)$ and $C_z = H(P(z)) - S(\rho)$, where in the CSS gauge the von Neumann entropy is easily given as $S(\rho) = L - N_x - N_y - N_z$. Without loss of generality we focus finding the Shannon entropy $H_x = H(P(x))$ using the above Lemma 4, and counting the number of uncertain measurements for a sequence of measurements $A_i = X_i$. To do this, we prove there exists a permutation of the sequence of measurements, $\{X_i\} \to \{X_{J(i)}\}$ such that measurements of the $J(i) = 1 \ldots (N_z + N_y)$ bits fall into case 4) in the proof for Lemma 4; the measurements of the $J(i) = N_z + N_y + 1 \ldots L - N_x$ bits fall into case 3); and the rest have zero uncertainty in the measurement outcome (case 1 or 2). Given such a result, we have $n_u = L - N_x = H_x$ and $C_x = N_y + N_z$.

Such a sequence can be found by Gaussian eliminating the columns $L+1 \ldots 2L$ of the rows $N_x + 1 \ldots L - S(\rho)$ of the stabilizer check matrix in the CSS gauge, such that the check matrix has the from in Eq. A12, but with $g_2^z$ in upper triangular form. If we take $J(i)$, for $i = 1 \ldots N_z + N_y$ to be the left most site for which the generator $i + N_x$ has a $Z_{J(i)}$ Pauli operator in it ( min $J(i)$ such that $(g_2^z)_{i,J} = 1$), we will ensure that $[X_{J(i)}, Z_{J(i)}] \neq 0$ for the measurements $i = 1 \ldots N_z + N_y$, and that they of case 4) above.

After this first sequence $N_z + N_y$ measurements, the stabilizer group $S_{i=N_z+N_y}$ will only contain operators that contains $X$ Pauli operators, and since the previous case 4) measurements don't change the number of independent generators we have $S(\rho_{i=N_z+N_y}) = S(\rho_{i=0})$. Thus, the remaining measurements fall into case 1, 2 and 3 outlined in the proof of Lemma 4. The measurements that fall into case 1 and 2 don't change the state or the entropy, and so we can choose the measurements $J(i)$ for $i = (N_z + N_y) \ldots N_z + N_y + S(\rho)$ to be the measurement that falls into case 3, such that the state loses one bit of entropy after each measurement $S(\rho_{i+1}) = S(\rho_i) - 1$. After this set of $S(\rho)$ measurements we will then have $S(\rho_i) = 0$, and the state as an $X$ basis state such that all subsequent measurement will have zero uncertainty in the outcome. Thus we have found the sequence $J(i)$ we set out to. In the case of pure states, we have $C_x = N_y + N_z = L - N_x$ and $C_z = N_y + N_x = L - N_z$. $\square$

### 4. Coherence free stabilizer states

In section III C, we claimed that a stabilizer state with zero coherence in the $X$ basis has the form

$$\rho_S(n) = \frac{1}{2^{k_n}} \sum_x |x\rangle \langle x| \prod_{i=1}^{L-k_n} \delta \left( \sum_j H_{ij}^x(n) x_j \right)$$

where $k_n = S(\rho)$ and $H_{ij}^x$ is the $k_n$ by $L$ matrix defining the generators of the stabilizer state $g_i^x = \prod_j X_j^{H_{ij}^x}$. This equality follows first from Theorem 6, which shows that such a state, which has $N_y = N_z = 0$ has zero coherence in the $X$ basis, $C_x = 0$. This implies that the density matrix $\rho_S(n)$ is diagonal $X$ basis such that it can be written as

$$\rho_S(n) = \sum_x |x\rangle \langle x| P(x) \qquad \text{(A14)}$$

Finally, we have that

$$
\begin{aligned}
P(x) &= \langle x| \rho_S(n) |x\rangle \qquad\qquad \text{(A15)} \\
&= \prod_{i=1}^{L-S(\rho)} \frac{1 + \langle x| g_i |x\rangle}{2} \\
&= \prod_{i=1}^{L-S(\rho)} \frac{1 + (-1)^{\sum_j H_{ij}^x x_j}}{2} \\
&= 2^{S(\rho)-L} \prod_{i=1}^{L-S(\rho)} \delta \left( \sum_j H_{ij}^x x_j \right)
\end{aligned}
$$

## Appendix B: Coherence Requirement for Alice

In this section we prove Theorem 2 used in the main text to describe Alice's coherence requirements to maintain a finite channel capacity.

**Theorem 2.** *Given a local Pauli basis D, any stabilizer mixed state $\rho$ with entropy $S(\rho)$ and relative entropy of coherence $C_D(\rho)$, there exists a sequence of $M > C_D(\rho)$ local Pauli measurements that reduce the entropy of $\rho$ (the state after measurement, $\rho'$, has entropy $S(\rho') < S(\rho)$).*

*Proof:* Without loss of generality, take the local Pauli basis $D$, to be the logical basis for the logical operators $X_i$. The proof proceeds by construction, and uses both the rules for stabilizer measurement given in section A 2, and the state represented in the CSS gauge given by Lemma 5, to identify the $M$ measurements that reduce the entropy of state $\rho$. By Theorem 6, the CSS gauge for the state $\rho$ has $N_y + N_z = C_x(\rho)$ and $N_x = L - S(\rho) - C_x(\rho)$. Preforming Gaussian elimination on the matrix

$$\begin{bmatrix} G_z^z \\ G_y^z \end{bmatrix} \qquad \text{(B1)}$$

will give the row operations to transform the $N_y + N_z$ generators, $\{g_i^z\} \cup \{g_i^y\}$ to have a unique left most site $k_i$ such that only one of the generators in this set has a Pauli operator $Z_{k_i}$ at site $k_i$ ($[g_i, X_{k_i}] \neq 0$). To guarantee some of the $M - C_x(\rho)$ measurements reduce the entropy of $\rho$, choose $C_x = N_y + N_z$ of the measurements to be of the operator $X_{k_i}$ on these unique left most sites. According to the rules in section A 2, the entropy of the state will remained the same after these measurements, but those $N_y + N_z$ generators will be replaced with the $X_{k_i}$ operators, such that state following those measurements will have $N'_x = L - S(\rho)$. If $M - C_x < S(\rho)$, then the remaining measurements, can be chosen to be a set of $\{X_i\}$ that can not be represented by the $N'_x$ generators ($X_i \in C(S)$ and $X_i \notin S$). According to the rules in section A 2, the entropy of state will be reduced by $M - C_x$. In the case $S(\rho) \leq M - C_x$, $S(\rho)$ of the measurements can be chosen in the same way, but now result in a purified state such that the remaining $M - C_x - S(\rho)$ of the measurements don't change the purity of the state $\square$.

## Appendix C: Proof of the coherence bound on the code distance

The coherence bound on the code distance for stabilizer codes is proven by making use of two properties of stabilizer states. The first is the lemma 4, which states the distribution of bit strings, $P(s)$ for given Pauli basis is uniform over $2^{H(P(s))}$ allowed bit strings ($P(s) = 1/2^{H(P(s))}$ if $s$ is allowed or $P(s) = 0$). The second useful property of stabilizer states is

**Lemma 7.** *Given a local Pauli basis $D$, any Pauli stabilizer state $|\psi\rangle$ can be reduced to a product state in $M = C(|\psi\rangle, D)$ measurements.*

*Proof:* Without loss of generality, take the local Pauli basis $D$, to be the logical basis for the logical operators $X_i$. Now imagine applying each measurement operator $X_i$ in order from $i = 1$ to $i = L$. Since the state $|\psi_i\rangle$ after measurement of $X_{i-1}$ is a stabilizer state, either $X_i |\psi_i\rangle = \pm |\psi_i\rangle$ and the measurement outcome is certain, or the measurement outcome is $\pm 1$ with probability $1/2$ and the measurement is completely uncertain. After all measurements are performed the state is in a product state, but the measurements whose outcomes were certain, did not need to be made as they didn't affect the state. Thus only the number of uncertain measurements $n_u(s) = M$ are needed to reduce the state to a product state. From Lemma 4, $n_u = H(P(s))$, and since the state $|\psi\rangle$ is a pure state, we have $n_u = C(|\psi\rangle, D)$ $\square$.

Treating such a sequence of $M$ measurements as an error on a state $|\psi\rangle$ encoding a set of logical qubits, we can then prove the desired theorem:

**Theorem 3.** *Given a local Pauli basis $D$, the code distance $d$ of a $[[N, k, d]]$ stabilizer code, $P$, is bounded by the coherence of the maximally coherent stabilizer state*

*in the code space:*

$$d \leq \max_{\psi \in P} C(|\psi\rangle, D) \equiv C_{PD} \qquad (C1)$$

*Proof:* Without loss of generality choose $D = X$ as the basis diagonal with respect to the Pauli $\{X_i\}$ operators. Then choose a complete set of logical Pauli operators $\tilde{Z}_n$ and $\tilde{X}_n$ acting on the code space, such that the basis states of the code, $|\psi_n\rangle$, diagonal with the logical $\tilde{Z}_n$ operators, contain the maximally coherent stabilizer state $|\psi_1\rangle$ (i.e. $C(|\psi_1\rangle, X) \geq C(|\psi\rangle, X)$ for all stabilizer states $|\psi\rangle$ in the code space). From lemma 7, the states $|\psi_n\rangle$ can be reduced to a product state in the computational basis with at most $C_{PD}$ measurements. Thus, there exists a projector $P_{s_n}$, for each basis state $|\psi_n\rangle$, with weight $C_{x,n} = C_x(|\psi_n\rangle) \leq C_{PD}$, that reduces that basis state $|\psi_n\rangle$ to a $X$ basis state in $C_{x,n}$ measurements: $P_{s_n} |\psi_n\rangle = 2^{-C_{x,n}/2} |x_n, s_n\rangle$ where $x_n$ are the value of the bits not projected by $P_{s_n}$ and $s_n$ are the values of the $C_{x,n}$ bits specified by the projector.

We now prove that one of the $P_{s_n}$, which has weight $M \leq C_{PD}$, must be an error and thus $d = M \leq C(P, D)$. We use proof by contradiction and assume all $P_{s_n}$ are correctable. This implies the error correction condition [72] for all $P_{s_n} \equiv P_n$:

$$\langle \psi_i | P_n P_m | \psi_j \rangle = \alpha_{nm} \delta_{ij}. \qquad (C2)$$

This condition for $n = m = 1$, such that $P_1$ is the $X$ basis projector associated to the maximum coherent stabilizer state $|\psi_1\rangle$, implies that all basis states must have the same coherence $C_{PD}$. First choosing $i = 1$, the condition implies $\alpha_{11} = \langle \psi_1 | P_1^2 | \psi_1 \rangle = \langle \psi_1 | P_1 | \psi_1 \rangle = P(s_1) = 2^{-C_{PD}}$ from lemma 4. For $i \neq 1$ the condition $2^{-C_{PD}} = \langle \psi_i | P_1 P_1 | \psi_i \rangle$ implies $P_1 |\psi_i\rangle = 2^{-C_{PD}/2} |\psi'_i\rangle$ where $|\psi'_i\rangle$ is a stabilizer state normalized to 1. Then from lemma 4, either $|\psi'_i\rangle$ is an $X$ basis state and $|\psi_i\rangle$ has coherence $C_{PD}$, or $|\psi'_i\rangle$ has some finite coherence $C' > 0$ such that $C_{x,i} = C_{PD} + C' > C_{PD}$. The second option is not valid from the assumption that $C_{PD}$ is the coherence of the maximal coherent stabilizer state, and so all basis states $|\psi_i\rangle$ must have coherence $C_{x,i} = C_{PD}$. Since $|\psi'_i\rangle$ must have zero coherence, the projector $P_1 \equiv P_{s_1}$, projects all basis states $|\psi_n\rangle$ to a product state $|x_n, s_1\rangle$.

Furthermore, from the error correction condition at $n = m = 1$ we have $\langle x_i, s_1 | x_j, s_2 \rangle = \delta_{ij}$ such that the states $P_1 |\psi_i\rangle$ are all orthogonal to each other. Now choose $|\psi_2\rangle$ to be the state obtained by flipping $\tilde{Z}_1$ of the first logical bit for the state $|\psi_1\rangle$, and consider the stabilizer state state $|+\rangle = (|\psi_1\rangle + |\psi_2\rangle)/\sqrt{2}$ obtained by applying a logical Hadamard gate to the first logical bit. Projecting $|+\rangle$ by $P_1$ gives us a stabilizer state $P_1 |+\rangle = (|x_1, s_1\rangle + |x_2, s_1\rangle)/2^{(C_{PD}+1)/2}$ which has coherence 1 because of the required orthogonality between $|x_1, s_1\rangle$ and $|x_2, s_1\rangle$. But this implies $|+\rangle$ has coherence $C_{PD} + 1$ which is a contradiction with the assumption $|\psi_1\rangle$ is the maximally coherent stabilizer state. Thus the error correction condition can not hold for all $P_n$ and the code distance $d = C_{x,m} \leq C_{PD}$. $\square$

## Appendix D: Measurement induced Markovian dynamics of coherence

### 1. Markovian dynamics of coherence in measurement-only circuits

In the main text, we discussed that the dynamics of coherence in the measurement-only limit are Markovian, and that they are described by the number of qubits polarized in the $X$, $Z$, and $Y$ directions, $N_x$, $N_z$, and $N_y = L - N_x - N_z$ respectively. The Markov chain is defined by the conditional probabilities $P(N_x(n), N_z(n)|N_x(n-1), N_z(n-1))$ for $N_x$ and $N_z$ at step $n$ given them at step $n - 1$. To determine the conditional probabilities, first consider the event of an $X_i$ measurement. If the measurement is made on a site polarized in the $X$ direction, the state does not change, but if it is made on a site polarized in $Y$ direction, the $X$ direction is learned and $Y$ direction is forgotten: $N_x \to N_x + 1$ and $N_y \to N_y - 1$. Given that a $X_i$ measurement is made, the probability this occurs is $N_y/L$. A similar thing happens if a $Z$ polarized bit is measured, which occurs with a probability $N_z/L$, thus we have:

$$P\left(N_x + 1, N_z | N_x, N_z\right) = p_x \frac{N_y}{L} \qquad \text{(D1)}$$

$$P\left(N_x + 1, N_z - 1 | N_x, N_z\right) = p_x \frac{N_z}{L}$$

$$P\left(N_x, N_z + 1 | N_x, N_z\right) = p_z \frac{N_y}{L}$$

$$P\left(N_x - 1, N_z + 1 | N_x, N_z\right) = p_z \frac{N_x}{L}$$

$$P\left(N_x - 1, N_z | N_x, N_z\right) = p_y \frac{N_x}{L}$$

$$P\left(N_x, N_z - 1 | N_x, N_z\right) = p_y \frac{N_z}{L}.$$

for the non-zero conditional probabilities. Using these conditional probabilities, a rate equation can then be derived for the average density of $X$ polarized qubits after $m$ measurements $\overline{N}_x(m) = \sum_{N_x} N_x P(N_x(m))$ as

$$\partial_m \overline{N}_x(m) = p_x \frac{L - \overline{N}_x}{L} - (p_z + p_y) \frac{\overline{N}_x}{L}, \qquad \text{(D2)}$$

with similar equations for $\overline{N}_y$ and $\overline{N}_z$. The steady state solution to these dynamics predicts the average steady state density of $\alpha$ polarized qubits is equal to $p_\alpha$ as intuitively expected: $\overline{N}_\alpha = p_\alpha L$ or equivalently for the coherences $C_\alpha = (1 - p_\alpha)L$.

### 2. Coherence dynamics in weak measurement limit

In this section we derive the conditional probabilities for the Markov process in the weak measurement limit

$p_m \sim O(1/L^2)$, arguing why, in this limit, each measurement has an uncertain outcome and changes the state. We again make use of stabilizer state tools, and in particular the CSS gauge of a stabilizer state discussed in appendix A 3. As discussed there, this representation of the stabilizer state has two parity check matrices $G_x^x$ and $G_z^z$ with $N^x$ and $N^z$ rows respectively. The generators specified by these check matrices, are all strings of all $X(Z)$ Pauli operators and therefore constrain $N^{x(z)}$ bits of information about the $X(Z)$ basis states that make up the stabilizer state. Theorem 6 then shows that the $X(Z)$ coherence of a stabilizer pure state is equal to the number of bits not known about the $X(Z)$ basis ($C_{x(z)} = L - N_{x(z)}$). Using this theorem, we can therefore focus on the conditional probabilities for the number of bits of information specified about the $X$ and $Z$ basis ($N_x$ and $N_z$) instead of the coherences directly. Notice that this is a generalization of the procedure for the measurement-only limit where the number of bits known about the $X(Z)$ basis states is equal to the number of physical qubits polarized in the $X(Z)$ basis.

We are therefore interested in obtaining the conditional probabilities $P(N_x(n), N_z(n)|N_x(n-1), N_z(n-1))$, and can determine them by determining the effect of an $X_i$ measurement on site $i$. The measurement can either act trivially on the state, if $|\psi\rangle$ is an eigenstate of $X_i$ (i.e. $[X_i, g_j] = 0$ for all generators $g_j$), or it can change the state (i.e. $[X_i, g_j] \neq 0$ for at least one $g_j$). To determine the probability that a measurement of $X_i$ acts trivially on the state, we first note that after $n = O(L^2)$ random CNOTs, each generator $g_j$ will have Pauli operators randomly distributed across the whole system. In the CSS gauge, the $g^x$ generators all commute with $X_i$, while the $g^z$ and $g^y$ generators have, after $n = O(L^2)$ random CNOTs, an equal probability of containing the $Z_i$ Pauli operator. Therefore we estimate the probability that the measurement of $X_i$ changes the state is $\approx 1 - (1/2)^{L-N_x}$, and approaches 1 in the thermodynamic limit as long as the coherence $C_x = L - N_x$ is $O(L)$.

When $[X_i, g_j] \neq 0$ for some $g_j$, the measurement of $X_i$ changes the stabilizer state and we must identify how the stabilizer state is updated. If the measurement outcome is $x_i$, then the updated state will obey $(-1)^{x_i} X_i |\psi\rangle = |\psi\rangle$, and we find that $(-1)^{x_i} X_i$ is a new generator of the stabilizer state with $N_x \to N_x + 1$. Then, to ensure all $g_j$ are commuting such that the state is a valid stabilizer state, we must, as described in appendix A 2, first change the representation of the stabilizer state so only one $g_k$ is non-commuting, and then remove it from the set of generators. The net effect is to replace the non commuting generator $g_k$ with the measurement operator $(-1)^{x_i} X_i$. Such an update can only be performed while at the same time maintaining the CSS gauge if one of the $\{g_j^z\}$ generators is chosen to be replaced (See Appendix D 4 for why). This results in $N_z \to N_z - 1$, and reflects the fact, that after the measurement, the qubit $i$ is in a superposition state of $Z_i$ basis states such that the coherence in the $Z$ basis,

$C_z = L - N_z$, has increased by one bit. Overall, we find that the $X_i$ measurement on a maximally entangled state increases the number of bits known about the $X$ basis states by 1 and decreases the number bits known about the $Z$ basis states by 1: $N_x \rightarrow N_x + 1$ and $N_z \rightarrow N_z - 1$. The effects of $Y_i$ and $Z_i$ measurements are determined similarly (See Appendix D 4), and we find that the non-zero conditional probabilities for the Markov chain are given as:

$$P(N_x + 1, N_z - 1 | N_x, N_z) = p_x \qquad \text{(D3)}$$
$$P(N_x - 1, N_z + 1 | N_x, N_z) = p_z$$
$$P(N_x - 1, N_z - 1 | N_x, N_z) = p_y$$

for $N_x$ and $N_z$ away from the Markov chain's boundaries: $N_x \geq 0$, $N_z \geq 0$ and $N_x + N_z \leq L$. The second boundary condition is because the number of generators allowed in the stabilizer state can not exceed $L$. The transition rates at the boundary are similarly determined and are shown in Fig. 7. We can therefore consider the dynamics of $(N_x, N_z)$ as a two dimensional random walk and derive the diffusion equation for the evolution of $P(N_x, N_z, m) = P(x, z, m)$ as

$$\partial_m P = (p_y + p_z - p_x)\partial_x \ P \qquad \text{(D4)}$$
$$+ (p_y + p_x - p_z)\partial_z \ P$$
$$+ 1/2(\partial_x^2 + \partial_z^2) \ P$$
$$- (p_x + p_z - p_y)\partial_x\partial_z \ P$$

which has drift velocity $(p_y + p_z - p_x)\hat{x} + (p_y + p_x - p_z)\hat{z}$ leading to the rate Eq. 11 discussed in the main text. Notice that special care must be taken on the $N_z = 0$ (and $N_x = 0$ by symmetry) boundaries. At $N_z = 0$ boundary the conditional probabilities are given as $P(N_x + 1, 0 | N_x, 0) = p_x$ and $P(N_x - 1, 0 | N_x, 0) = p_y$ and result in the diffusion equation

$$\partial_m P(z=0) = \left( (p_y + p_z - p_x)\,\partial_x + \frac{1}{2}\partial_x^2 \right) P(z=0).$$

The steady state solution on this boundary gives the localization length $\lambda \sim 1/(p_y + p_z - p_x)$ in the $x$ direction as discussed in the main text.

### 3. Finite measurement rate dynamics

The two limits discussed above offer solutions for the steady state coherence in two extremes: 1) $p_m/p_u \rightarrow \infty$, in which the probability that a measurement of $X_i$ is uncertain depends on the number of bits known about the $X$ basis; and 2) $p_m/p_u \rightarrow 1/L^2$ in which the probability of an uncertain measurement outcome depends only on the rates $p_x$, $p_z$ and $p_y$. When $p_x > p_y + p_z$ these two extremes are distinguished by volume-law v.s. area-law coherence in the $X$ basis. This suggest the possibility of a coherence transition as a function of measurement rate $p_m$. This possibility is ruled out by the Clifford simulations shown in the top panel of Fig. 13, which only

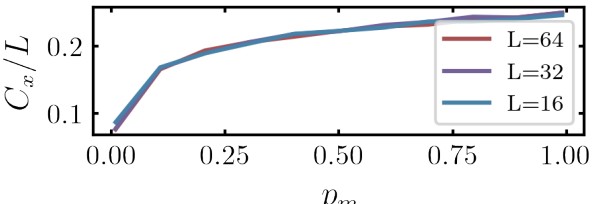

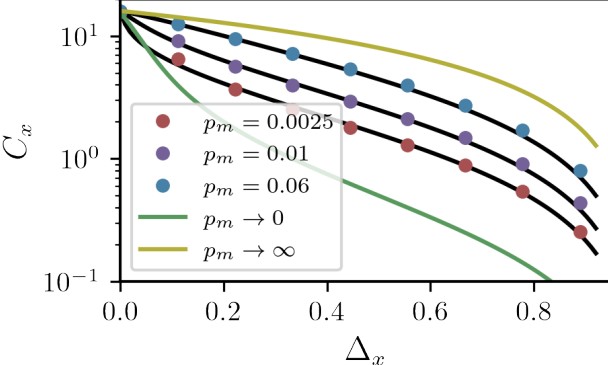

Figure 13. Steady state coherence $C_x$ as a function of $p_m$ and $\Delta_x = (p_x - p_z)$ for $p_y = p_R = p_e = 0$. The top figure shows volume-law coherence $C_x$ for $\Delta_x = 0.5$ obtained via Clifford simulations. It shows no evidence of a phase with area-law coherence for finite $p_m$. The bottom figure shows the steady state coherence $C_x$ for different values of $p_m$ and a system size $L = 32$. The colored dots are data computed via Clifford simulations, while the solid lines correspond to the predictions using the coherence rate equations derived in the text. The three black lines correspond to the rate equation Eq. 12 where the lengths scales $\xi = (2.7, 5, 8)$ are found by best fit for the Clifford simulation data at $p_m = (0.06, 0.01, 0.0025)$ respectively.

shows volume-law coherence. Thus, the weak measurement limit, $p_m < 1/L^2$ is a finite size effect and does not exist as $L \rightarrow \infty$ for finite $p_m$.

To access the finite measurement rate limit, we first note that the two extremes discussed above correspond to two distinct structures of the generators in the late time stabilizer states: 1) when $p_m/p_u \rightarrow \infty$ and $g_i$ are single site Pauli operators, and 2) when $p_m/p_u \rightarrow 1/L^2$ and the generators $g_i$ have extent scaling with system size. To interpolate between these two limits and access the finite $p_m$ dynamics, we make the assumption that the generators of the stabilizer state have instead a finite extent $\xi$ and are centered at sites evenly spaced throughout the chain. We proceed as before and determine the probability that the measurement of $X_i$ changes the state and the coherence. This occurs if one of the generators $g_i$ does not commute with $X_i$, which under the above assumption, is only possible for generators centered at most $\xi$ sites away. If we take $\beta_x$ as the probability one of these generators commutes with $X_i$, then the probability all generators commute with support on site $i$ is $\beta_x^\xi$. Thus the probability a measurement of $X_i$ is uncer-

tain and obtains information about the $X$ basis is $1 - \beta_x^\xi$, yielding the rate equation

$$\partial_m \overline{N}_x = p_x(1 - \beta_x^\xi) - p_z(1 - \beta_z^\xi) - p_y(1 - \beta_y^\xi) \quad \text{(D5)}$$

In the weak measurement limit, we expect $\xi \sim L \to \infty$, since the generators are assumed to have extent over the entire system. This is consistent with the fact that in the limit $\xi \to \infty$, Eq. D5 reproduces the weak measurement rate equation Eq. 11 for $p_y = 0$. If instead we work in the measurement-only limit, the stabilizer state becomes a product state with $\xi = 1$. If we choose $\beta_x = \frac{\overline{N}_x}{L}$, $\beta_x = \frac{\overline{N}_z}{L}$ and $\beta_y = \frac{L - \overline{N}_x - \overline{N}_z}{L}$, then the rate equation Eq. D5 will have the form of Eq. D2. Therefore, we have a single phenomenological parameter $\xi$ to interpolate between the two extreme limits of strong and weak measurement. The steady state coherence for a given $\xi$ is then given by the following implicit equation $\partial_m \overline{N}_x = 0$, which can be solved numerically. Numerical solutions for $C_x = L - \overline{N}_x$ are shown in the bottom panel of Fig. 13, and agree well with Clifford simulations of $C_x$ when $\Delta_x > 0.1$ for a single choice of the length scale $\xi$.

### 4. Markov Chain Effects of $Z$ and $Y$ measurements

Above, in section D 2, we presented the conditional probabilities, Eq. D3 for the Markov chain in the weak measurement limit and derived the contribution from the $X_i$ measurements. That derivation relied on the fact that the measurement of $X_i$ can only be performed on a stabilizer state while maintaining the CSS gauge if one of the $\{g_j^z\}$ generators is used to perform gaussian elimination. This is seen as follows. First, all generators which do not commute with $X_i$ contain a $Z_i$ Pauli operator. Since there are generally $O(L)$ $\{g_j^z\}$ operators, with stabilizer extent $L$, at least $C_z > 1$ of them is likely to contain $Z_i$. This is also true of the $\{g_j^y\}$ generators, but if one of them, say $g_k^y$, is used to perform the gaussian elimination of $Z_i$, the $N_z$ $\{g_i^z\}$ generators will all contain the $X$ Pauli string of the $g_k^y$ operator used for elimination. After this procedure, $g_y^x$ will then contain $C_z - 1$ linear dependent rows and the state will not be in the CSS gauge. This does not occur if one of the $\{g_j^z\}$ generators is uses for elimination.

The derivation of the contribution from $Z$ measurements is exactly the same as the first due to the duality between $X$ and $Z$ measurements in the stabilizer gauge. The contribution from $Y$ measurements occurs because it becomes, with high probability, a generator in $\{g_j^x\}$ and another in $\{g_j^z\}$ will not commute with the $Y_i$ measurement in the $L \to \infty$ limit. Thus one generator (say the one in $\{g_j^x\}$), will have to eliminate the other ($\{g_j^z\}$) leading to $N_z \to N_z - 1$ and $N_y \to N_y + 1$. Then the row in $g_x^x$ used for elimination will be replaced with $Y_i$ leading to $N_x \to N_x - 1$ and $N_y \to N_y + 1$. Thus for a $Y$ measurement $N_x$ and $N_z$ both decrease by 1 as in the third equation in Eq. D3.

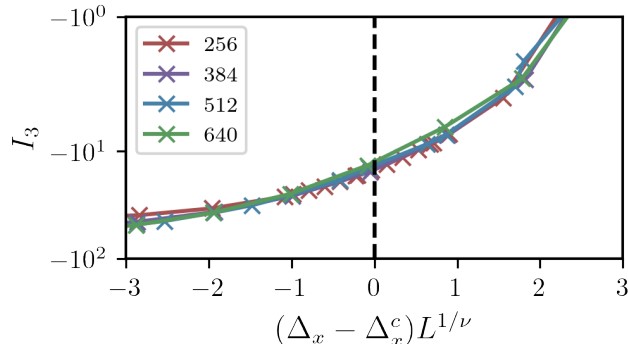

Figure 14. Scaling collapse for tripartite mutual information $I_3$. Data for the different system sizes cross at $\Delta_x = \Delta_x^c = 0.333 \pm 0.005$ confirming the $\Delta_x = 1/3$ critical point predicted by the dynamics of coherence. The curves collapse for $\nu = 1.2 \pm 0.05$ where the error source is sampling error from the finite, $O(2000)$, circuit realizations performed which we estimate to be $\Delta I_3 \approx 0.5$. Here we do not show lines for $L \leq 128$ owing to non-universal finite size effects causing a slight drift in the apparent critical point. For example, the curves for $L = 128$ and $L = 256$ cross at $\Delta_x^c = 0.35$.

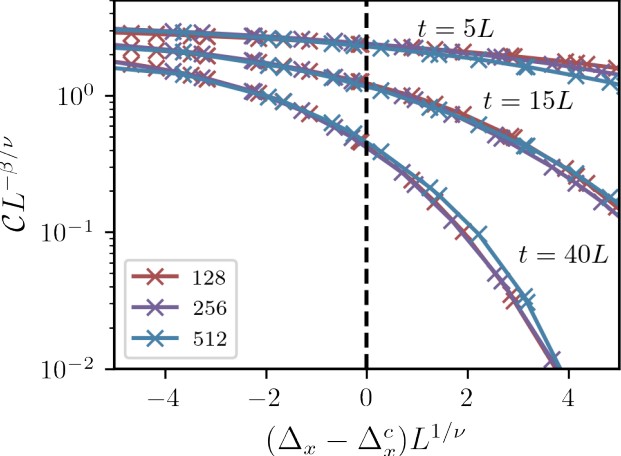

Figure 15. Scaling collapse for the coherent information, $\mathcal{C}$, occurring at three different times $t = 5L, 15L$ and $t/L = 40$ as labeled in the figure. In this figure $\Delta_x^c = 0.33 \pm 0.02$, $\nu = 1.09 \pm 0.05$ and $\beta = 0.65 \pm 0.05$, where the variation in the exponents arises from both sampling error as in Fig. 14 and from variations in the best fit critical parameters at the three different times. The length critical exponent, $\nu$ is compatible by a single standard deviation with the one obtained in Fig. 14 for the tripartite mutual information.

### Appendix E: Critical properties of the coherence controlled entanglement transition

In section IV, we presented a phase transition controlled by the relative rate of $X, Y$ and $Z$ measurements of a circuit composed of CNOTs and measurements at a fixed overall measurement rate $p_m = 0.01$. The phase

transition was observable in the half cut entanglement entropy at late times, $S(L/2)$, the biparitite, $I_2$, and tripartite, $I_3$, mutual informations and the coherent information, $\mathcal{C}$ between the initial and final state of the system. These quantities identified a critical point of $\Delta_x^c = (p_x - p_z)/(1 - p_y) = 1/3$ when $p_y = 1/4$, and $p_x + p_z = 1 - p_y$. In this appendix, we determine the length critical exponent, $\xi \sim (\Delta_x - \Delta_x^c)^{-\nu}$ and the exponent $\beta$ for the coherent information, $\mathcal{C} \sim (\Delta_x - \Delta_x^c)^\beta$.

To identify these critical exponents, we make the following scaling hypothesis

$$
\begin{aligned}
I_3(\Delta_x) &= f\left((\Delta_x - \Delta_x^c)\,L^{-\nu}\right) \\
\mathcal{C}(\Delta_x) &= L^{\beta/\nu} g\left((\Delta_x - \Delta_x^c)\,L^{-\nu}\right),
\end{aligned}
\tag{E1}
$$

and find that data from our numerical simulations confirms these hypothesises in Fig. 14 and Fig. 15. Under this scaling hypothesis, $I_e$ and $\mathcal{C}$ collapse to a single polynomial function of $p$ for different $L$ and $t$. The critical parameters are determined by optimizing a fit to this polynomial. This gives the critical parameters, $\Delta_x^c = 0.33 \pm 0.02$, $\nu = 1.09 \pm 0.05$ and $\beta = 0.65 \pm 0.05$

with an optimal residual of $10^{-2}$ for the tripartite mutual information $I_e$, and an optimal residual of $10^{-3}$ for the coherent information $\mathcal{C}$. These exponents are distinct from the critical exponents found for the transition described in Ref. [26, 98], where the transition is controlled by a competition between measurements and unitaries. This is in contrast to the current setting in which the transition is controlled by a competition between coherence generating measurements and coherence destroying generating measurements. In particular, Ref. [26] finds the coherent information exponent as $\beta = 0$, which is not compatible with the critical properties we observe in the present scenario.

The critical exponent $\nu = 1.09$ is consistent with directed percolation which describes the classical transition discussed in section III and in Refs. [75, 76]. Future work could find it interesting to better understand if the transition is in the directed percolation universality class or not. An obstacle to this is apparent large finite size effects occurring in these circuits as discussed in the caption of Fig. 14.

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
