# Peer review of "Coherence requirements for quantum communication from hybrid circuit dynamics"

_SciPost Physics_

## Round 2 · Referee Report · Anonymous (Referee 1) · 2023-8-28

Strengths

The paper makes a connection between measurement induced & purification transitions, and the amount of quantum coherence provided in to the state (whether initially or during its evolution). This connection is new and interesting. It may lead to further insight into this timely subject.

The paper is very well written and presents a deep a discussion into the role of coherence. For example, it nicely distinguishes between classical and quantum information and shows that coherence is essential to protect the latter from errors, while the former can be protected even in its absence.

The paper also makes a nice connection between coherence resources and the ability to perform quantum error correction.

Weaknesses

I don't see many weaknesses. The paper is a bit long and possibly a table of contents would be helpful. But overall it flows very nicely.

Report

The authors investigate the role of quantum coherence in the ability to protect against attacks in a quantum channel. They show that a purification transition can be tuned by tuning the amount of coherence and thus emphasize the role of coherence in the measurement phase transition. The paper also elucidates the very nicely the difference between classical and quantum channels and what resources are needed to protect information in each case. By that it makes a connection to quantum error correction codes and their breakdown.

The paper is original, interesting and important. It may lead to further insight into the field of quantum information and the measurement phase transition in general. I therefore recommend to publish this paper.

I have a few humble question:

1. The coherence transition is tuned, eg, by tuning the ratio of Y measurements vs. Z and X at very small pm. It seems to me that if pm is large enough there is a second transition into a "disentangling" phase even if py>|px-pz|, right? Therefore, is it possible that the transition at py=|px-pz| is not the same as the MPT at large pm? (also, I understand the special role of Y is set by the basis choice for the CNOT gate?) This connects to the question whether the difference in the critical exponents compared to Ref.[26] (Appendix E) is indeed due to the difference in the applied gates or because the coherence transition discussed here is different in nature? The first explanation sounds suspicious to me and I would recommend to check it unless I am wrong and there is no second transition.

2. The authors comment about gates drawn from the random Haar measure. A numerical verification of this statement might a nice compliment to the paper (using the amount of coherence they produce to predicts the location of the coherence transition). However, I understand this is beyond the scope of this paper?

  • validity: top
  • significance: top
  • originality: top
  • clarity: top
  • formatting: perfect
  • grammar: perfect

Author:  Shane Kelly  on 2023-10-27  [id 4074]

(in reply to Report 1 on 2023-08-28)
Category:
remark
answer to question

Dear Referee, Editor and Readers,

We thank the referee for their careful reading, their comments, and their positive assessment of our manuscript. We have considered the comments by the referee, clarified the relevant points in our manuscript and responded to the their two questions below. We hope that the editor and the referee will find our work suitable for publication.

Yours Sincerely, Shane P. Kelly, Ulrich Poschinger, Ferdinand Schmidt-Kaler, Matthew P.A. Fisher, Jamir Marino

Question 1

With regards to the referee’s first question: Yes, the basis for the CNOT gate sets the special role of the Y basis. As chosen, the CNOT preserves coherence in both the X and Z basis, but not the Y basis. Also, yes there is a second transition for $p_y>|p_x-p_z|$ tuned by the measurement rate. We added a sentence to the main text to highlight this second point.

The two transitions are tuned by two different processes, and our numerics suggest the scaling features are distinct. In the first case, at small $p_m$, the transition results from a competition of coherence generating (Y measurements) and coherence destroying processes (X or Z measurements). In this case, the length scaling exponent $\nu$, found from numerics, suggest the transition is in the percolation universality class. This is similar to another case of competing non commuting measurement processes [1]. In contrast, the transition tuned by $p_m$ is tuned by a competition between unitary and measurement dynamics typical considered [2] which does not show percolation like exponents. This seems the most plausible explanation for the apparent difference in the scaling exponents for the two transitions.

Nonetheless, we hesitate to definitively claim that the two transitions are of a distinct universality, and leave open this problem for future investigations. Our main reservation is that at small $p_m$, finite size effects tend to be exaggerated. Furthermore we know that for vanishing measurement rate, $p_m=O(L^{-2})$, the transition is completely controlled by the Markovian coherence dynamics shown in Fig 7. In this limit, $O(L^2)$ CNOT gates occurring between measurements maximizes entanglement under the coherence bound on entanglement. Furthermore the dynamics of coherence follow a simple stochastic process shown in Fig 7, yielding scale free features at the transition similar to an unbiased random walk. We do not see scaling behavior predicted by this limit, but the small $p_m=0.01$ in which we see the coherence tuned transition may still be effected by the simple $p_m=1/L^2$ limit for the system sizes we have access too. Thus, we leave open the question of if there are two distinct universality classes in the infinite size limit.

Question 2

A numerical simulation for Haar gates is beyond the scope of this work, but we have clarified our expectations for this type of dynamics. We also now emphasized that this is a direction for future work.

References

[1] Phys. Rev. Lett. 127, 235701 (2021) [2] Phys. Rev. B 101, 060301 (2020).

---

## Round 2 · Referee Report · Anonymous (Referee 2) · 2023-9-10

Strengths

1 - new connection between measurement induced transitions and coherence properties, with applications to quantum channels

2 - extensive discussions and carefully written

Weaknesses

the only minor weakness I could spot is that the discussion is at time too qualitative, and one has to resort to consulting appendices often

Report

In their manuscript, Kelly and coworkers discuss the dynamics of (random) Clifford circuits in relation to coherence. The main goal is to achieve an understanding of channel properties under a number of scenarios, where Alice and Eve are allowed only certain types of operations.

Relevant cases discussed are those in which case Alice protects a classical diary (without inputing coherence/entanglement into the system), the competition between quantum phases in case both Alice and Eve can perform coherence-enhancing operations, and quantum communication channels. While the results have overall only somewhat distant implications for concrete quantum communication tasks, their conceptual insights are important. In addition, the work sheds light on the overall connection between measurement induced transitions and quantum communication protocols in the context of coherence, something that could easily see further applications.

Overall, the paper is well written, and its conclusions are sound and interesting.

Presentation-wise, I found the authors' choice of deferring to the appendices most technical details somewhat unfortunate, as I had to stop reading to understand what was really done at least a few time. This is of course a matter of personal taste, and I am not insisting on substantial changes: just, I'd recommend the authors to consider moving some of the material back to the text (maybe, shortening the discussion paragraphs at the beginning of some subsections, like V.B, etc.)

I have append below some more detailed comments:

1) why is 'random' crossed in the caption of Fig. 1?

2) Sec. III.a has a somewhat bizzare order. I'd anticipate the content of III.a.1 before conclusions are drawn (in fact, I was unable to understand how Fig. 5 was obtained when it is discussed, this is only discussed in III.a.1);

3) I have missed the discussion about error bar averaging. While the number of realizations considered is large (as are system sizes), I still find hard to believe that late time entropies (see, e.g., Fig. 4) have negligible error bars.

4) Frankly, I did not find particularly surprising the fact that "Alice can not encode quantum information without being able to generate X coherence". Isn't this somehow to be expected? Now, I understand that this then requires details to be nailed down properly - as the authors do in Sec. III. But still, I'd take emphasis out of this claim, and just mention this is a sanity check that illustrates the relevance of coherence in these settings.

About appendices:

A1) the collapse scaling in Fig. 14 is not clear. It would be informative to discuss the quality of the collapse by reporting, e.g., the sum of residuals.

A2) in appendix E, since (as correctly pointed out by the authors) the circuits considered here are quite different from [26], I think the sentence:

In particular, Ref. [26] finds the coherent information exponent as β = 0, which is inconsistent with our data.

shall be changed into something like

"In particular, Ref. [26] finds the coherent information exponent as β = 0, which is not compatible with the critical properties we observe in the present scenario."

Requested changes

please see the questions above, some of which require revisions.

  • validity: high
  • significance: high
  • originality: top
  • clarity: good
  • formatting: reasonable
  • grammar: perfect

Author:  Shane Kelly  on 2023-10-27  [id 4075]

(in reply to Report 2 on 2023-09-10)

Dear Referee, Editor and Readers,

We thank the referee for their careful reading, their comments, and their positive assessment of our manuscript. We have considered the comments by the referee, clarified the relevant points in our manuscript and responded to the their remarks below. We hope that the editor and the referee will find our work suitable for publication.

Yours Sincerely, Shane P. Kelly, Ulrich Poschinger, Ferdinand Schmidt-Kaler, Matthew P.A. Fisher, Jamir Marino

Remark 2

We thank the referee for identifying this issue. Our intention in using this order of presentation is to separate the discussion of the classical mutual information from the discussion of purification dynamics. In doing so, we forgot to specify the distribution of initial bits $P_0(x_0)$ used to evaluate the classical mutual information between initial and finial bits $x_0$ and$ x_n$. Having clarified this, we believe Fig. 5 can now be understandable before section III.a.1

Remark 3

The reason for small error bars is that, for the brickwork random circuits considered, the late time entropies have circuit to circuit statistical fluctuations that are suppressed in system size. This has been discussed both in unitary and hybrid circuit dynamics, where the late time entropies are mapped to extensive observables of an effective statistical mechanics model. In particular, the entropies map to the corresponding free energy of the statical mechanics model.

We have added the following sentence to clarify in the main text:

“In that figure, and in numerical results presented through out the manuscript, statistical fluctuations due to circuit sampling are suppressed in system size. This is a generic feature of entanglement growth[1, 2] in random circuit models.”

Remark 4

The referee is right that this result isn’t particularly interesting, and we ensured our introduction and abstract do not highlight it. In particular we changed

“We first show that in the limit of zero coherence, Alice can only protect classical information.” to “We first consider the limit of zero coherence, and confirm Alice can only protect classical information.”

Remark A2

We thank the referee for the suggestion, and have added to the appendix the following sentences

“Under these scaling hypothesis, the $I_e$ and $\mathcal{C}$ collapse to a single polynomial function of $p$ for different $L$ and $t$. The critical parameters are determined by optimizing a fit to this polynomial. This gives the critical parameters, $\Delta_x^c=0.33\pm0.02$, $\nu=1.09\pm0.05$ and $\beta=0.65\pm0.05$ with an optimal residual of $10^{-2}$ for the tripartite mutual information $I_e$, and an optimal residual of $10^{-3}$ for the coherent information $\mathcal{C}$. “

Referees concerns on presentation

We have taken referees suggestion A2, and removed the typo on the crossed out word in Fig 1. We acknowledge the referees preference to include more of the appendix in the main text, but have kept the organization of the manuscript unchanged. This is because the topics in the appendix are either not central to the main message or are to technical. In particular, appendices A-D require knowledge of stabilizer states, which we assume the general reader may lack.

References

[1] Phys. Rev. X 7, 031016 (2017) [2] Phys. Rev. B 101, 104301 (2020)

---

## Round 2 · Referee Report · Anonymous (Referee 3) · 2023-10-4

Strengths

Well-written manuscript discussing timely topics (error-correcting codes, monitored dynamics) from an unusual, but important perspective (coherence resource theory).

Weaknesses

Some technical aspects deserve further comments, as well as the generality claims in the manuscript conclusion

Report

Dear Editor,

The work by S. Kelly and collaborators discusses coherence (as a resource theory) in a class of stabilizer circuits, encapsulating aspects of measurement-induced dynamics and error-correcting codes.
In particular, the Authors consider three variant of a quantum game (with adaptive Clifford circuits) from a classical setup to a classical setup with quantum initial states to a fully quantum setting to corroborate their results, namely:
1. The determination of the coherence requirements for quantum communications in stabilizer circuits.
2. Coherences shed light on what is genuinely quantum about measurement-induced transitions, corroborating previous discussions in [75-77]. (Remark: the Authors should add Rev. Lett. 129, 260603 to the references [75-77] as this manuscript is relevant to the topic).

Overall, the manuscript is well written and presented, and the pedagogical structure may help inexperienced readers. Therefore, the manuscript is worthy of publication in Scipost Physics once the Authors discuss the following comments.

As highlighted by the Authors, basis dependence is intrinsically related to the resource theory of coherence. Indeed, the observable of choice, i.e., the relative entropy of coherence, includes the participation entropy.
It is, therefore, a natural question: are there basis independent results? For instance, it is known that participation entropy captures the dynamical criticality, e.g., Ref. [53] and [SciPost Phys. 15, 045 (2023)]. While the leading term is volume, the subleading term captures the universal basis-independent content of the phase transition. Given the close analogy between participation entropy and the relative entropy of coherence, it would be interesting to clarify this point. Can the Authors comment on this point?

Relatedly, I'm not sure that going beyond stabilizer setups will be a mild variation (referring to Sec. VI A). Indeed, suppose I consider a non-Pauli basis, like a random (local or Global) Hilbert space basis. In that case, any well-defined stabilizer state will be fully delocalized; hence, the coherence relative entropy may be uninformative.
Therefore, it is non-trivial to generalize the ideas and results of this paper to more general setups, including Haar circuits or measurements on non-Pauli strings: magic, for instance, can play a critical role.
Therefore, I suggest the Authors highlight these non-trivial interplay (e.g., beyond Pauli basis dependence, global stabilizer basis-dependence, etc.) in their conclusions and soften the generality section of their results (if they don't have quantitative data supporting these conjectures).
(Ultimately, this is good news: coherences in more generic setups are an interesting venue for future investigations!)

  • validity: high
  • significance: good
  • originality: high
  • clarity: high
  • formatting: excellent
  • grammar: excellent

Author:  Shane Kelly  on 2023-10-27  [id 4076]

(in reply to Report 3 on 2023-10-04)

Dear Referee, Editor, and Readers,

We thank the referee for their careful reading, their comments, and their positive assessment of our manuscript. We have considered the comments by the referee, clarified the relevant points in our manuscript, and responded to their two questions below. We hope that the editor and the referee will find our work suitable for publication.

Yours Sincerely, Shane P. Kelly, Ulrich Poschinger, Ferdinand Schmidt-Kaler, Matthew P.A. Fisher, Jamir Marino

Response to referee’s questions

Referee’s Question 1

As highlighted by the Authors, basis dependence is intrinsically related to the resource theory of coherence. Indeed, the observable of choice, i.e., the relative entropy of coherence, includes the participation entropy. It is, therefore, a natural question: are there basis-independent results? For instance, it is known that participation entropy captures the dynamical criticality, e.g., Ref. [53] and [SciPost Phys. 15, 045 (2023)]. While the leading term is volume, the subleading term captures the universal basis-independent content of the phase transition. Given the close analogy between participation entropy and the relative entropy of coherence, it would be interesting to clarify this point. Can the Authors comment on this point?

Response

In our work, while the von-Neumann entanglement (as shown in Fig. 9) captures basis-independent correlations and criticality, the relative entropy of coherence does not appear to. For example, while figure 10 shows the coherence in the X basis crosses $L/2$ at the critical point, the coherence in the Z basis (not shown) remains large. We suspect the difference between this work and the results shown in SciPost Phys. 15, 045 (2023) is that while the participation ratio measures real space localization in the Anderson transition, coherence does not measure if the dynamics is localized in real space in our work. In fact, even when the coherence is zero, as discussed in section 3, the purely classical dynamics spread classical information across the qubits. Another difference is that even in the localized phase, the unitary dynamics of the Anderson model are capable of increasing coherence, even if only by a constant amount scaling with the localization length. In contrast, the CNOT unitary gates investigated here cannot increase coherence at all. The relation between Anderson localization and the MIPT (both tuned by measurements and coherence-destroying operations) is an interesting direction for future work.

Referee’s Question 2

Relatedly, I'm not sure that going beyond stabilizer setups will be a mild variation (referring to Sec. VI A). Indeed, suppose I consider a non-Pauli basis, like a random (local or global) Hilbert space basis. In that case, any well-defined stabilizer state will be fully delocalized; hence, the coherence relative entropy may be uninformative. Therefore, it is non-trivial to generalize the ideas and results of this paper to more general setups, including Haar circuits or measurements on non-Pauli strings: magic, for instance, can play a critical role. Therefore, I suggest the Authors highlight these non-trivial interplays (e.g., beyond Pauli basis dependence, global stabilizer basis dependence, etc.) in their conclusions and soften the generality section of their results (if they don't have quantitative data supporting these conjectures). (Ultimately, this is good news: coherences in more generic setups are an interesting venue for future investigations!)

Response

The referee highlights an important point about the generalization of our work. We now emphasize in the discussion that all of our results apply only to the coherence in a local basis. Notice that the difference between a local Pauli basis and any local basis is simply a matter of the definition of the Pauli operators. What is important is that the operations applied are Clifford operations with respect to a fixed definition of Pauli operators. In this regard, we have also clarified and softened our expectations of generality in section VI A. In particular, we only expect similar results to hold if the coherence-generating operation is replaced by a non-Clifford gate. As the referee points out, if the dynamics are dominated by a generic unitary, coherence in a local basis would not be relevant. Similarly, if the CNOT gate was replaced by a non-Clifford gate, it would generate coherence in the local Pauli basis, and the dynamics would not be constrained by the coherence in the X or Z basis as in the main text. We also expanded more on why we expect Theorem 2 to generalize for non-stabilizer states and proposed a possible route toward generalizing it.

---

## Editorial Decision

resubmitted